# Summertime tropospheric ozone source apportionment study in the Madrid region (Spain)

David de la Paz[1], Rafael Borge[1], Juan Manuel de Andrés[1], Luis Tovar[1], Golam Sarwar[2], Sergey L. Napelenok[2]

[1]Laboratory of Environmental Modelling, Department of Chemical & Environmental Engineering, Universidad Politécnica de Madrid, (UPM), c/ José Gutiérrez Abascal 2, 28006, Madrid, Spain

[2]Center for Environmental Measurement & Modeling, U.S. Environmental Protection Agency, Research Triangle Park, NC, USA

*Correspondence to*: Rafael Borge (rafael.borge@upm.es)

**Abstract.** The design of emission abatement measures to effectivly reduce high ground-level ozone ($O_3$) concentrations in urban areas is very complex. In addition to the strongly non-linear chemistry of this secondary pollutant, precursors can be released by a variety of sources in different regions and locally produced $O_3$ is mixed with that transported from the regional or continental scales. All of these processes depend also on the specific meteorological conditons and topography of the study area. Consequently, high-resolution comprehensive modeling tools are needed to understand the drivers of photochemical pollution and to assess the potential of local strategies to reduce adverse impacts from high tropospheric $O_3$ levels. In this study, we apply the Integrated Source Apportionment Method (ISAM) implemented in the Community Multiscale Air Quality (CMAQv5.3.2) model to investigate the origin of summertime $O_3$ in the Madrid region (Spain). Consistent with previous studies, our results confirm that $O_3$ levels are dominated by non-local contributions, representing around 70% of mean values across the region. Nonetheless, precursors emitted by local sources, mainly road traffic, play a more important role during $O_3$ peaks, with contributions as high as 25 ppb. The potential impact of local measures is higher under unfavorable meteorological conditions associated with regional accumulation patterns. These findings suggest that this modeling system may be used in the future to simulate the potential outcomes of specific emission abatement measures to prevent high-$O_3$ episodes in the Madrid metropolitan area.

## 1. Introduction

Air pollution is one of the main environmental problems and is recognized as a global threat to public health. In 2019, 4.2 million people died prematurely worldwide as a result of a poor air quality (WHO, 2021). Even in regions that have taken decisive actions to curb emissions, such as Europe, over 300,000 premature deaths (EU27) are currently associated to air pollution, most of them related to high levels of $PM_{2.5}$ (particles with aerodynamic diameter of $\leq$ 2.5 microns) (238,000) and $NO_2$ (nitrogen dioxide) (49,000) (EEA, 2022). In recent years, concentrations of many of the regulated pollutants in Europe have decreased as a result of a general reduction of emissions. From 2009 to 2018, the concentration of $PM_{10}$ (particles with aerodynamic diameter of $\leq$ 10 microns), $PM_{2.5}$ and $NO_2$ diminished on average by 19%, 22% and 18-23% (depending on the air quality monitoring station type), respectively (EEA, 2020). These measures, however, have not reported comparable reductions of ozone ($O_3$) ambient concentration levels.

Tropospheric $O_3$ is a secondary pollutant formed from photochemical reactions between many different precursors, mainly nitrogen oxides ($NO_x$ = NO (nitric oxide) + $NO_2$) and non-methane volatile organic compounds (VOCs) (Seinfeld and Pandis, 2016; Jenkin and Clemitshaw, 2000; Monks et al., 2015). According to the last European Union (EU) emission inventory

report (EEA, 2022), the most important activity sectors regarding $O_3$ precursors emissions are the "Road transport" sector (7% and 37% of total VOCs and $NO_X$ emissions, respectively), the "Commercial, institutional and households" sector (15% and 14%, respectively) and the "Solvent and product use" sector, representing 42% of total VOCs emissions. Once emitted from urban and industrial areas, these precursors are subsequently transported by the prevailing wind regime (Xu et al., 2011). Atmospheric life-time of $O_3$ depends on numerous variables. In the boundary layer, atmospheric life-time of $O_3$ is short, roughly 1 or 2 days, depending on the abundance of precursors (Young et al., 2013). In the free troposphere, its lifetime can be of up to 2 weeks, time enough to be transported long distances, from the local to the global scale (Monks et al., 2015; Stevenson et al., 2006). In addition to in-situ formation, transport of $O_3$ from the stratosphere is also relevant to explain the tropospheric ozone levels (IPCC, 2007; Hsu et al., 2005). Furthermore, this gas exchange between layers of the atmosphere is expected to increase in the future globally (Meul et al., 2018; Banerjee et al., 2016) due to dynamic and chemical changes in the atmosphere induced by climate change.

Due to these complex dynamics, tropospheric $O_3$ levels have not decreased (Jung et al., 2022; Sicard et al., 2023) in accordance to significant $NO_X$ and VOCs emissions reduction (45% and 41%, respectively in the 2009- 2018 period). As a result, 12% of the urban population in Europe is still exposed to high $O_3$ concentrations according to EU regulations, with a toll of 24,000 premature annual deaths (EEA, 2022), especially in the Mediterranean basin (Amann, 2008; EEA, 2018, EEA, 2020). The share of urban population that suffers from excessive exposure to $O_3$ rises to 95% (EEA, 2022) when the World Health Organization (WHO) guidelines are considered (WHO, 2021). Of note, tropospheric $O_3$ produces both short-term (Bates et al., 1972; Bell et al., 2004; Goodman et al., 2018) and long-term health effects (Jerrett et al., 2009; Seltzer et al., 2018), impacting the population living in large urban agglomerations as well as their surroundings. Moreover, it also may have relevant effects on ecosystems (De Andrés et al., 2012; Mills et al., 2011; Harmens et al., 2011) and climate (Sitch et al., 2007; Stocker et al., 2013; IPCC, 2015).

Globally, the latest studies using satellite data suggest that tropospheric $O_3$ average levels increased over the last four decades (Ziemke et al., 2019; Gaudel et al., 2018). Paoletti et al. (2014) evaluated observations from monitoring stations in the United States (US) and Europe from 1990 to 2010 and concluded that the $O_3$ annual average increased by 7%/year in rural stations and around 12-17%/year (US and EU, respectively) in urban stations. However, $O_3$ formation is highly non-linear and trends may change depending on the evaluated time period and region, the metric used, and other local factors such as topography or the proximity to the precursor's emission sources (Reche et al., 2018; Massagué et al., 2023). According to specific studies for the Iberian Peninsula, the trend of the annual average of $O_3$ for rural stations in the 2004-2012 period was not clear (Querol et al., 2014). In contrast, an increasing trend around $1 - 3$%/year was observed in all seasons in urban, traffic and industrial stations. Borge et al. (2019) reported an average increase of $10\,\mu g \cdot m^{-3}$ of daily 8-hour maximum $O_3$ moving average concentrations (MDA8) for the 1993-2017 period. However, they detected that the highest increase related to fall and winter months (up to $19\,\mu g \cdot m^{-3}$), in agreement with general increases of the oxidation capacity in the atmosphere of the largest urban areas in Europe modeled by Jung et al. (2022).

Nonetheless, the $O_3$-forming photochemical activity is largely regulated by weather conditions, especially temperature and solar radiation. For this reason, tropospheric $O_3$ formation has a marked seasonal character, with the highest $O_3$ values typically recorded in spring and summer (Logan, 1985; Granados-Muñoz and Leblanc, 2016), especially in those locations that are highly influenced by nearby urban areas (Brodin et al., 2010; Carnero et al., 2010) where large amounts of precursors are emitted. Therefore, understanding summertime $O_3$ dynamics is more relevant from air quality management perspective.

Furthermore, information on the relative importance of emission sources on ambient levels should be considered when designing plans and measures, especially when they target highly non-linear secondary pollutants such as $O_3$ (Cohan and Napelenok, 2011).

There are different source apportionment techniques that may support air pollution research and decision making (Thunis et al., 2019). Approaches based on sensitivities, such as single-perturbation or brute force methods (Borge et al., 2014, Tagaris et al., 2014, Zhang et al., 2022, Qu et al., 2023) may be useful to anticipate the potential effect of a given intervention. However, tagging methods (Grewe et al., 2017, Butler et al., 2018) provide fully mass conservative apportionment at receptors of interest and may be better suited for diagnosis purposes (Borge, 2022). These pollution tracking capabilities have been integrated into modern air quality models to provide attribution information together with the standard concentration and deposition output fields, can be successfully applied to study pollution dynamics (Simon et al., 2018; Pay et al., 2019, Li et al., 2022). This approach may be particularly interesting to describe how $O_3$ levels are linked to emission sources under unfavorable meteorological conditions (Cao et al., 2022; Zohdirad et al., 2022) or specific local atmospheric circulation patterns (Zhang et al., 2023) that may lead to high concentration events (Lupaşcu et al., 2022).

This research focuses on the center of the Iberian Peninsula, encompassing the city of Madrid and its surroundings. Consistently with general emission trends in Europe, the emission of the main $O_3$ precursors in the Madrid region decreased by 47%, for VOCs, and by 44% for $NO_X$ from 1990 to 2018 (CM, 2021). While recent control measures succeeded in reducing $NO_2$ levels (AM, 2022), such emissions reductions have, at the same time, substantially impacted urban atmospheric chemistry by modifying its oxidative capacity. Recent studies (Saiz-Lopez et al., 2017; Querol et al., 2016) suggest that $O_3$ levels have increased in Madrid by 30-40% during the 2007-2014. A greater decrease in NO emissions than in $NO_2$ emissions (with the subsequent reduction of the $NO/NO_2$ ratio) may be one of the factors responsible for this response (Querol et al., 2016; Querol et al., 2017; Zaveri et al., 2003; Jhun et al., 2015). The exceedances of the target value for the protection of human health in the region mainly occur in summer periods, especially under adverse meteorological conditions that have been extensively characterized in previous studies (Querol et al., 2016; Querol et al., 2017; Millan et al., 2000; Plaza et al., 1997; Querol et al., 2018; Pay et al., 2019; Escudero et al., 2019). Preventing these exceedances in the region requires an understanding of the source attribution of $O_3$, specially under specific weather patterns that may lead to high pollution levels (Zhang et al., 2023).

In this research, we apply a state-of-the-science air quality model to provide insights into the emission sources and transport patterns which are involved in the formation of tropospheric $O_3$ during typical summertime conditions in the Madrid region. In addition to contributing to the scientific understanding of photochemical pollution, the final purpose of this work is to inform the decision-making process needed to design further emission reduction measures in the study area.

## 2. Methodology

### 2.1. Modeling system

The research is supported by a mesoscale modeling system with three main components. Meteorological fields are generated by WRFv3.7.1 (Weather Research and Forecasting) (Skamarock and Klemp, 2008). Physics options and parameterizations (Table S1 in the supplement) are based on previous studies (Borge et al., 2008a; de la Paz et al., 2016) and WRF outputs were postprocessed with MCIP v5.1 (Meteorology - Chemistry Interface Processor) (Otte and Pleim, 2010). Emission processing relies on the US EPA SMOKEv3.6.5 (Sparse Matrix Operator Kernel System) model (Institute and Environment, 2015; Baek

and Seppanen, 2018) that has been specifically adapted for the Iberian Peninsula (Borge et al., 2008b; Borge et al., 2014). Biogenic emissions are generated by MEGAN v2.1 (Model Emissions Gases and Aerosols from Nature) (Guenther, 2006; Guenther et al., 2012). The third component is the CMAQv5.3.2 (Community Multiscale Air Quality) modeling system (Byun and Schere, 2006; Ching and Byun, 1999). This 3D chemical-transport model (CTM) simultaneously predicts the concentration of all relevant substances considering transport (advection and diffusion), chemical transformation and deposition. Gas-phase atmospheric chemistry is represented by the Carbon Bond 6 (CB06) (Yarwood et al., 2010) chemical mechanism with chlorine chemistry (CB06r3) (Sarwar et al., 2012; Whitten et al., 2010, Emery et al., 2015) according to SPECIATE 4.0 (Hsu et al., 2006) while the module AERO6 (Appel et al., 2013) is used to describe aerosol dynamics and chemistry. Considering the influence of different scales, from the continental to the regional-urban, on $O_3$ levels (Valverde et al., 2016; Pay et al., 2019; Baker et al., 2016; Han et al., 2018), boundary conditions are of particular interest. Previous studies in the Iberian Peninsula have demonstrated that $O_3$ is particularly sensitive to boundary conditions (Borge et al., 2010). For a more realistic representation of the boundary influence, the mother domain receives 1 hour-resolution, dynamic chemical boundary conditions from hemispheric CMAQ (Mathur et al., 2017) simulations.

In this study, the Integrated Source Apportionment Method (ISAM) (Kwok et al., 2013) implemented in CMAQv5.3.2 (Napelenok, 2020; Shu et al., 2023) is used. ISAM provides apportionment capability of the full concentration and deposition output arrays including the gaseous photochemically active species such as $O_3$ as well as inorganic and organic particulate matter. The CMAQ-ISAM implementation used in this study attributes source identity to secondary pollutants based strictly on reaction stoichiometry with all reactions playing a role that are relevant to the formation and destruction of any species in the chemical mechanism. ISAM is highly customizable for any number of user-specified combinations of emissions source sector and geographical source areas. For $O_3$, this implementation differs from the previous ISAM versions (including CMAQv5.0.2) that attribute the formation of secondary pollutants to source sectors based on chemical regime – NOx- or VOC-limited $O_3$ formation (Kwok et al., 2015) and from other studies where precursor attribution is directed by the user to either NOx or VOC emissions, such as Butler et al. (2020). Regime-based methods are useful to attribute secondary species that depend on multiple precursors. However, the regime determination relies on predefined thresholds of different metrics, often the $H_2O_2/HNO_3$ ratio (Sillman, 1995) that dynamically depend on location and time specific parameters (Li et al., 2022). By strictly following stoichiometry of all chemical reactions in the mechanism, this version of ISAM avoids the necessity to make decisions and assumption regarding ozone formation regimes. Decisions on tagging method selections are highly dependent on the specific application and the scientific and/or regulatory aims of each individual study. As the needs of the scientific and regulatory communities evolve, so do the apportionment methodologies. Since the conclusion of this study, CMAQ-ISAM has been expanded to include the regime-based, the stoichiometry-based, as well as other configuration options. More information on ISAM as well sample application and comparison results can be found in Shu et al. (2023).

## 2.2. Modeling domains

The three nested domains shown in Figure 1 were used to perform numerical simulations in this study. This layout is intended to capture medium (Millán et al., 1991) and long-range influences of $O_3$ transport (Zhang et al., 2020; Qu et al., 2021; Brook et al., 2013) and to provide enough resolution over the area of interest to depict local dynamics (Plaza et al., 1997; Borge et al., 2022). The mother domain (D1) includes Europe and Northern Africa with a 12 km x 12 km spatial resolution while D2 is centered over the Iberian Peninsula and has a 4 km x 4 km spatial resolution (Table S2 in supplement). The innermost domain (D3) used in this study covers Madrid and surrounding areas with 1 km² spatial resolution (136 km in the east-west direction

and 144 km in the north-south direction). All three domains have a common 35-level vertical structure covering the whole
Troposphere with 18 layers within the first kilometer to accurately represent atmospheric processes within the planetary
boundary layer (Borge et al., 2010).

The region has a continental Mediterranean climate with an annual mean temperature of 14.6 ºC and 367 mm of accumulated
precipitation with a typical summer drought (https://www.madrid.org/iestadis/fijas/coyuntu/otros/cltempe.htm). The Central
Range (Sierra de Guadarrama), with maximum elevations of 2500 meters above sea level (m.a.s.l.), crosses the D3 modeling
domain in the NE-SW direction and divides it into two main regions; the northern and southern plateaus of the Iberian
Peninsula. The southern half of the domain, where the city of Madrid (with an average elevation of 657 m) is located, features
the Tajo river basin. This topography configures a dominant wind circulation along the NE-SW direction and enhances
anticyclonic stagnation conditions (Plaza et al., 1997; Querol et al., 2018) usually induced by the semi-permanent Azores High
(García et al., 2002). $O_3$ formation typically peaks with high temperature and solar radiation under stagnation conditions
(Querol et al., 2018; Reche et al., 2018; Garrido-Pérez et al., 2020) that often occur at summertime.

## 2.3. Temporal domain

Model simulations were completed for July 2016, using a previous 3-day period as model spin-up. According to the Spanish
Meteorological Agency (AEMET, 2017) it was an unusually warm month (with an average temperature of 25.5 ºC), being the
4[th] hottest month of July since 1961 in the Iberian Peninsula. It was also a dry month, with 13% less precipitation than the
average of the month in the 1981-2010 reference period. Considering the meteorological trends in this region (Borge et al.,
2019), it may be considered as a representative summer period for modern weather conditions. More importantly, this period
was selected because of an intensive experimental campaign carried out to characterize ozone episodes in Madrid and
surroundings (Reche et al., 2018). This period was thoroughly analyzed by (Querol et al., 2018) that identified two typical
circulation patterns associated to venting and accumulation episodes. The later are characterized by weak wind forcing (wind
speed <4-5 m s$^{-1}$), stable conditions and air stagnation that favor $O_3$ local formation. Oppositely, stronger winds (> 7 m s$^{-1}$)
promote advection and prevents from reaching $O_3$ peaks under venting conditions.

During this period (2016), 26 out of the 42 air quality monitoring stations in the innermost (D3) modeling domain (Figure 1),
recorded exceedances of the concentration threshold related to the $O_3$ target value for the protection of human health (MDA8
> 120 µg·m$^{-3}$). The highest number of exceedances (up to 359 in the month, 47% of total annual exceedances) were found
around the Madrid metropolitan area, in the city outskirts. Of note, no exceedances of the MDA8 were recorded downtown
Madrid.

## 2.4.    Emission sources for the apportionment analysis

Emissions for this modeling exercise result from the combination of the official national (MMA, 2018), regional (CM, 2021)
and Madrid's city local inventory (AM, 2021). These inventories are compiled according to the EMEP/EEA standardized
methodology (EEA, 2019) and are conveniently adapted, spatio-temporally resolved for modeling purposes (Borge et al.,
2008b; Borge et al., 2018) and consistently combined for the different modeling domains (Borge et al., 2014).

Emissions from power generation and industrial activities (SNAP 01, SNAP 03 and SNAP 04 groups according to the Selected
Nomenclature for Air Pollution nomenclature) were merged due to their limited presence in this modeling domain. Since
emissions from agriculture (SNAP 10) in the region are mainly significant for VOCs from plants, they have been tagged along

biogenic VOC (BVOC) emissions from vegetation (SNAP 11) (labeled as SNAP 10-11 in Figure 12). Soil-$NO_X$ emissions provided by MEGAN 2.1 (Yienger and Levy, 1995) are also included in this group and their share to total $NO_X$ emissions is around 4% in this period, consistently with MEGAN results reported European scale (Visser et al., 2019).

Consequently, 8 emission sources (Table 1) were tagged for the source apportionment analysis of ambient $O_3$ in the region. The share of $NO_X$ and VOC emissions of each of them for July 2016 is summarized in Figure 2. The emission breakdown on an annual basis can be found elsewhere (Borge et al., 2022). Figure 2 shows that they account for the totality of emissions in the modeling domain and identifies road traffic (SNAP 07) as the main source of $NO_X$ (66% of total emissions), followed by other mobile sources (SNAP 08). Since emissions from the residential, commercial and institutional sector (SNAP 02) occur almost exclusively in winter, the contribution from this sector is relatively small (around 7%) and is related to combustion units in agriculture and forestry. VOC emissions are dominated in this period by emissions from plants. The combined contribution of forests and agriculture represents 72% of total VOCs. Solvent use (SNAP 06) is the main anthropogenic source of this $O_3$ precursor with a total share of 22% (nearly 80% of anthropogenic VOC emissions).

In addition to the attribution of $O_3$ ambient levels to the emissions within the modeling domain, hereinafter referred to as local sources, the contribution of boundary conditions (BC) and initial conditions (IC) are also estimated in this study. Considering the typical $O_3$ daily patterns and the variability of circulation patterns, the latter refers to the initial mixing ratios on a daily (24 hour) basis, i.e., each day is run separately using the outputs from the previous day as IC. This is a difference with most previous source apportionment studies that analyze shorter periods (Pay et al., 2019) or specific high concentration events (Lupaşcu et al., 2022; Zhang et al., 2022). While this may hinder the comparability of our results, this methodological option may be appropriate considering the temporal span of the period analyzed (a whole month), the typical diurnal cycle of $O_3$ and the goal of characterizing this attribution under specific meteorological conditions. This helps understanding differences on $O_3$ source apportionment depending on regional circulation patterns (Zhang et al., 2023) and explicitly considering the influence of vertical transport of $O_3$ from residual layers form previous days that may lead to rapid increases of $O_3$ concentrations near the surface (Qu et al., 2023 and references within). Therefore, this approach may be better suited to provide useful information for decision making, especially for the design of short-term action plans intended to control ozone peaks.

## 3. Results

The results are presented in four subsections. Firstly, the main features of the simulated period and model performance are presented. Then, an overview of the source apportionment analysis carried out in the study area for the whole month is discussed. Finally, this same analysis is performed for two specific days representative of different circulation patterns defined by Querol et al. (2018): advective pattern (July 13th) and accumulation pattern (July 27th). Additional information for July 20[th] and July 6[th], identified by Querol et al. (2018) as advective and accumulation days, respectively, is provided in the supplement. Finally, the temporal patterns of the $O_3$ apportionment are examined at the location of the air quality monitoring stations within the simulation domain. Aggregated results by station type are discussed in 3.4 while the results for different geographical areas relative to the location of Madrid city (quadrants) are presented in the supplement.

## 3.1.    Ozone levels during the study period and model evaluation

While this period was hotter and dryer than most of recent summers, July 2016 may be representative of typical summer conditions in the Madrid region and included a concatenation of characteristic local circulation patterns (Plaza et al., 1997) with direct implications on ground-level $O_3$ (Querol et al., 2018; Escudero et al., 2019). Figure 3 presents both observed and modeled concentration series at representative points (Figure 1), and shows the venting and accumulation days identified in Querol et al. (2018). The time series demonstrate that $O_3$ levels are significantly lower under venting conditions, although significant differences are found depending on the location, which supports the need to use high-resolution modeling systems to analyze pollution dynamics in the Madrid region. On the other hand, accumulation patterns tend to produce higher concentrations (up to 175 µg/m³), especially during July 27th.

It can be observed that the model is able to reproduce the temporal patterns, as confirmed by the high correlation coefficients (r) and index of agreement (IOA) shown in Table 2. The statistical evaluation demonstrates a reasonable model performance, yielding better statistical results than recent simulation studies in this domain. Pay et al. (2019) reported an aggregated correlation coefficient of 0.66 and mean bias (MB) of 22.5 ug/m³ for the central region of the Iberian Peninsula. In this study, we obtained an average r value of 0.74 and a MB of 6.2 ug/m³. Of note, 95.2% and 66.7% of the r values for the locations of the 42 monitoring stations used in this study are larger than 0.6 and 0.7, respectively while the overall normalized mean bias (NMB) is only 9%. The results for a series of common statistics (Borge et al., 2010) for each of the monitoring sites in our modeling domain can be found in Table S3. The model, however, may have some difficulties capturing the amplitude of observed $O_3$ series and fails to accurately reproduce concentration peaks on some days. This is evidenced by the relatively large error in comparison with the bias (23% and 9%, respectively as an average over the 42 monitoring stations in the modeling domain). In the supplementary material (Table S4), we present a separate model performance assessment for accumulation and advective patterns showing that the main differences among them relate to errors, both MGE and RMSE that are systematically higher for accumulation periods. This may be related to the limitations of the meteorological model to depict atmospheric circulation during stagnant conditions suggested by Pay et al. (2019). Even when WRF was found to outperform other models for this particular episode (Escudero et al., 2019), the ability to reproduce wind direction and wind speed clearly deteriorates for accumulation periods, as shown in Table S5.

As expected, results are poorer for urban background and traffic locations, since the typical spatio-temporal representativeness of the measurements in such locations is not comparable with that of a mesoscale modeling system, even with 1 km² spatial resolution.

## 3.2.    Spatial analysis of the source apportionment assessment

In this section, we discuss the contribution to ground-level $O_3$ of the tagged sources (Table 1) both, for monthly average and high values (illustrated by the 90th percentile, hereinafter P90). $O_3$ apportionment focusses on anthropogenic sources since they have more interest from the point of view of possible abatement measures (Oliveira et al., 2023) and have a larger contribution than that of SNAP10-11 (below 4% to total $O_3$ levels in this period). However, it is not a negligible apportionment since these groups account for 27% (monthly mean) and 22% (P90) of total $O_3$ averaged over the Madrid region when BC and IC are not considered (Figure S1). Non-anthropogenic emissions have been reported to play an important role on atmospheric photochemistry and they interact with manmade emissions so, they need to be considered in the process of designing policies to reduce tropospheric $O_3$ levels. Therefore, we discuss the potential role of emissions from agriculture and nature as well.

### 3.2.1. Non-anthropogenic sources

According to our results, the combined contribution of SNAP 10-11 represents around 21% and 28% of that of local anthropogenic emissions to monthly mean and P90. This is a similar relative importance to that reported by Sartelet et al. (2012) at European scale. As well as Collet et al. (2018), they argue that the influence of BVOC becomes stronger on VOC-limited areas which is consistent with our findings (Figure S2) since the Madrid region is predominantly $NO_X$-limited in summer, except for the metropolitan area of Madrid city and surroundings, that remains VOC-limited all year round (Jung et al., 2022, Jung et al., 2023). Pay et al. (2019) did not quantify explicitly the contribution of biogenic emissions to ozone in the Iberian Peninsula. However, the contribution of "other", that included emissions from SNAP 11 along with other sectors was around 5% in the center of the Iberian Peninsula, even though biogenic emissions represent a large fraction of total VOCs.

The contribution of BVOC to ozone levels in Europe reported by Tagaris et al. (2014), Karamchandani et al. (2017) or Zohdirad et al. (2022) are slightly larger (below 6%) and are even more according to some source apportionment at global scale for this latitude (Grewe et al., 2017; Butler et al., 2020). It should be noted that different experimental design and apportionment algorithms would lead to significant differences (Zhang et al., 2017; Borge et al., 2022) preventing the direct comparison of the results from different studies. In addition to the apportionment methodology itself, the results may differ depending on the emission inventory used, the modeling scale and resolution, temporal span and sources tagging scheme. Nonetheless, the contribution of biogenic emissions found in our work is not remarkably different than those previously reported, especially for this same geographical area.

Previous studies suggested that relatively low contributions of biogenic VOCs to $O_3$ levels may relate to underestimations of isoprene levels (Lupașcu et al., 2022), a very relevant specie for $O_3$ chemistry (Dunker et al., 2016) that constitutes more than 25% of global biogenic VOC emissions Guenther et al. (2012). Nonetheless, it is widely recognized that BVOC emission estimates involve large uncertainties (Poupkou et al., 2010; Wang et al., 2017; Zhang et al., 2017) and the MEGAN model used in this study has been found to overestimate isoprene emissions (Wang et al., 2017 and references within). According to our inventory, isoprene represents 48% of total BVOC. While isoprene ambient measurements are not made routinely, Querol et al. (2018) recorded an average level of isoprene around 0.2 ppb in Majadahonda, a suburban site ~15 km away from downtown Madrid (in the west, northwest direction) between July 5th and July 19[th], 2016. That is in relatively good agreement with the results of CMAQ in our simulation, that predicted slightly less than 0.1 ppb for that location and period and reproduced quite accurately the average daily pattern (see Figure S3 in the supplementary material).

Arguably, the relatively low contribution of BVOC in our and previous studies in this area (Valverde et al., 2016; Pay et al., 2019) may be a consequence of the underestimation of isoprene mixing ratios. However, that is compatible with the stronger influence of other anthropogenic VOC species reported elsewhere. Querol et al. (2018) estimated the total ozone formation potential (OFM) applying the maximum incremental reactivity (MIR) proposed by Carter (2009) to the VOC measurements made in their campaign for the same period and location than our study. Based on this methodology, they identified formaldehyde as the single most important compound (35.5% of total OFP) while isoprene was ranked 7[th] with an OFP below 5%. By family, primary BVOCs represented 6% of total OFP as an average during the experimental campaigns in this period. Similar studies elsewhere (e.g. Meng et al., 2022 in the Pearl River Delta region) conclude as well that the ozone formation potential of BVOCS is lower than that of anthropogenic VOCs applying a similar reactivity scale (Carter and Atkinson, 1989). That may be consistent with the apparent insensitivity of $O_3$ to isoprene emissions reported in other studies (Simpson, 1995; Jiang et al., 2019; Ciccioli et al., 2023).

Of note, SNAP 10-11 include $NO_X$ emissions from soils (see section 2.4). Although they represent less than 4% of total $NO_X$ emissions in the domain, they may be underestimated by MEGAN (Visser et al., 2019). According to other studies, i.a. Weng et al. (2020), emissions from agricultural soils may be substantially higher and could pose a significant constrain towards the control of $O_3$ levels (Lu et al., 2021). Methods to reduce the uncertainty of $NO_X$ emissions estimates from soils as well as their role for $O_3$ control policies specifically for this region may be addressed in future research.

Other non-controllable sources include stratospheric ozone, also tagged in this study (ST in Figure 12). This source informs about the influence of vertical injections on ground level $O_3$ levels (Hsu et al., 2005) and the potential contribution reported in this region for specific extraordinary ozone levels (San José et al., 2005). Pay et al. (2019) hypothesize that stratosphere-troposphere exchange (STE) may have played a significant role towards the end of July 2016 in the Iberian Peninsula. According to our results, however, the direct transport of $O_3$ from the stratosphere in our modeling domain was negligible in this period, with 1-hour maximum contributions below 0.4 ppb in the southwest end of the Madrid region (see Figure S4). This contrasts with remarkably higher contributions reported in other areas of Europe (Lupaşcu et al., 2022) and those from global simulations for similar latitudes (Butler et al., 2018). It should be noted that here we account for $O_3$ STE exclusively within our innermost nested domain and part of the $O_3$ attributed to BC may be related to contributions from the Stratosphere in other regions.

### 3.2.2. Anthropogenic sources

Figure 4 shows the contribution to ground-level $O_3$ of the BC and that of all local anthropogenic emissions combined for both, monthly average and high values (illustrated by the 90[th] percentile, hereinafter P90). In both cases BC is the largest contributor. This is consistent with previous studies that have identified boundary conditions as the dominant contribution to ground-level $O_3$, i.a. Pay et al. (2019) for the Iberian Peninsula, Collet et al. (2018) for the USA or de la Paz et al. (2020) for Madrid specifically. However, the weight of each of the sources on both metrics is different. As an average, 70% of the mean $O_3$ levels in the Madrid region comes from BC (Figure 4a), while for P90, the contribution from BC is considerably smaller, around 50% (Figure 4b).

The maximum anthropogenic contribution for the monthly average (Figure 4c) reaches 17% (7.5 ppb in absolute terms), with a mean contribution of 8.7% over the whole Madrid Region (Figure S1). Regarding P90 (Figure 4d), the maximum contribution of local anthropogenic emissions is 28% (in the center and southwest of the Madrid municipality), around 22 ppb in absolute terms. This corresponds to a spatially averaged contribution of 12.2% over the Madrid (Figure S1), which corresponds to an absolute value around 11 ppb. Despite the general dominance of BC on $O_3$ levels, these results point out the relevance of local emissions (Figure 2) is higher for $O_3$ peaks, a consistent finding with those of previous studies (Valverde et al., 2016; Qu et al., 2023).

Figure 5 shows the apportionment to P90 of each emission sector for local sources. Consistently with Valverde et al. (2016) and Pay et al. (2019) our results clearly identify road transport (SNAP07) as the most influential sector, contributing 41% to P90 as an average over the Madrid region. The contribution of this sector (relative to local sources) reaches values up to 55% in the proximity of the main communication routes (Figure 5d). In absolute terms, this means an average contribution of 5 ppb and a maximum one of 11 ppb (Figure S5). The next sector with the highest contribution relates to off-road mobile sources (SNAP08), with an average contribution in the Madrid region of 17% (1.8 ppb) and a maximum of 8 ppb in the vicinity of the Adolfo S)uárez Madrid-Barajas airport. This suggests that $NO_X$ emissions play a more important role than VOC emissions in

the photochemical production of ozone, in concurrence with previous source apportionment studies (Dunker et al., 2016; Butler et al., 2018; de la Paz et al., 2020). Nonetheless, the importance of controlling anthropogenic VOC emissions to prevent high $O_3$ episodes has been noted in previous studies (Cao et al., 2022), even in regions with strong biogenic emissions (Coggon et al., 2021). In addition to the contribution of BVOC previously discussed, anthropogenic VOC had also an influence on $O_3$ levels during July 2016 in the Madrid region (see Figure S2). While the spatially-averaged attribution of SNAP 06 to P90 is only 1.5 ppb with maximum contributions of 3 ppb at specific locations (southwest of Madrid as shown in Figure S2 and Figure S5), emissions from the use of solvents and other products can reach values up to 20% of total anthropogenic contributions to $O_3$ P90 (Figure 5c). This is comparable to the contribution of all industrial sources combined (SNAP01-03-04). This may be related to the high OFP of aromatics within SNAP 06 VOC (Meng et al., 2022) and is consistent with the findings of Oliveira et al. (2024) that attributed 64% or total OFP to the solvent sector (relative to that of total anthropogenic VOC) in densely urbanized areas such as Madrid. Coggon et al. (2021) also found that consumer and industrial products (included in SNAP 06 group) are important precursors of ozone in urban areas, were typically present a VOC-sensitive regime. Nonetheless, they found that $O_3$ formation may take a few hours and the maximum contributions of VOC emitted in New York City to occur a few tens of km away, close to $NO_X$-limited areas. Our high-resolution analysis indicates that a similar process may take place in Madrid too. The rest of the sectors analyzed (SNAP05 and SNAP09) have negligible contributions (around 0.05 ppb as an average over the Madrid region).

If the analysis is done on a daily basis, it is worth noting the significance of the initial conditions (IC) as well, with a spatially-averaged contribution of 19% and of 34% to monthly average and P90 $O_3$ levels, respectively (Figure S1). However, the role of IC is more relevant to analyze how meteorological conditions may affect the source apportionment. Of note, in this study IC refers to $O_3$ from the previous 24-hour period. Consequently, the effect of IC on $O_3$ does not necessarily diminish throughout the month. Instead, we found that the influence of IC relates mainly to regional circulation patterns. We elaborate on this in the following sections.

### 3.3. Source apportionment assessment under characteristic circulation patterns

The study of the influence of meteorology on the $O_3$ ambient levels is carried out by analyzing the results for specific days representative of the two circulation patterns. Querol et al. (2018) identified an advective pattern for 13[th] and 20[th] July and an accumulation pattern for 6[th] and 27[th] July. In this section, we examine the source apportionment for those dates (13[th] and 27[th] in more detail) to test the hypothesis that local atmospheric conditions may induce a significant difference on $O_3$ attributions, as reported elsewhere (Zhang et al., 2023).

Figure 6 shows the daily average and the P90 of hourly $O_3$ levels during the accumulation and advective episodes. It is observed that during accumulation days (6[th] and 27[th]), mixing rations averaged over the Madrid region were 13 - 20% higher than those of advective periods (days 13[th] and 20[th]) and, although not shown, around 4 - 8% higher than the monthly average. Regarding the maxima, the average P90 (3[rd] highest hourly mixing ratio for a given day) during the accumulation periods in the Madrid region may be 25% higher than that of the ventilation periods.

### 3.3.1. Accumulation pattern

Consistent with previous studies that highlight the role of meteorology on $O_3$ (Nguyen et al., 2022), modeling results show that accumulation days are especially relevant regarding the potential impacts on health and vegetation and a deeper analysis

of pollution dynamics under those conditions is of interest. Figure 7 shows the hourly evolution (3:00, 9:00; 15:00, 21:00 UTC) of surface $O_3$ mixing ratios during the day 27th (day 6th is shown in Figure S7), along with $O_3$, NOx and VOCs vertical levels up to 5 km height for a NE-SW cross section, related to the dominant wind directions (the same results for a perpendicular SE-NW cross section are shown in Figure S8 in the supplement).

A low $O_3$ mixing ratio surface layer (around 40 ppb) can be clearly seen for early hours of the day (03:00 UTC, 05:00 local time). This relates to a shallow Planetary Boundary Layer (PBL) (a few hundred meters high) and weak winds from the NE (between 1-2 m s$^{-1}$). Around 6:00 UTC (08:00 local time), the main emitting sectors (such as road transport) begin to emit $O_3$ precursors (see Quaassdorff et al., 2016) for characteristic emission temporal profiles). The prevailing surface wind directs the urban plume towards the SW and the southern slope of the Sierra de Guadarrama (Figure S6). Of note, the wind direction aloft

is the opposite, in accordance with recirculation processes reported for this domain (Plaza et al., 1997). As the day progresses (09:00 UTC, 11:00 local time), the PBL height grows (up to 1.5 km) as radiation and temperature increase, mixing $O_3$ vertically. At the same time, the emissions of precursors (concentrated in the Madrid city, MD) lead to an increase in the local production of $O_3$ in the plume, more evidently in the rural areas (NOx limited regions) in the leeward side of the city. On the contrary, in the vicinity of high NO$_X$ emission intensity areas, $O_3$ is consumed by NO through the reaction NO+$O_3$ → NO$_2$ +

$O_2$, a titration effect documented in previous studies (Saiz-Lopez et at., 2017).

Over the following mid-day hours (09:00-15:00 UTC, 11:00-17:00 local time) the PBL further develops and a vertical homogenization process occurs. There is a deep vertical mixing of newly formed ozone with $O_3$-enriched upper layers generated in previous days (Querol et al., 2018; Escudero et al., 2019). As illustrated in Figure 8, there is a first $O_3$ reservoir located around 1500 m altitude (at 00:00 UTC, 02:00 local time) that relates mainly to local sources and contributes with 2-8

385 ppb, while higher $O_3$ reservoirs (around 4000 meters a.s.l) relate to BC and have a considerably higher contribution (50-75 ppb). Around 15:00 UTC (17:00 local time) the PBL reaches 3000 - 4000 m in accumulation periods and $O_3$ levels up to 75 - 80 ppb are found (Figure 7). This dynamic is compatible with the ozone sounding (https://woudc.org/data/explore.php) included in Figure 9, that shows a very constant $O_3$ value around 65 to 70 ppb from the surface to 4000 m a.s.l.

Later, around 17:00 UTC, the local $O_3$ production from anthropogenic local emissions released earlier is maximum (Figure 8),

with ground-level contributions that can reach 30 ppb SE in the municipality of Madrid. However, the greatest contribution during these hours continues to be from the BC (up to 50 - 60 ppb at surface level). From 21:00 UTC, the PBL has already decreased to a few hundred meters, the turbulence dwindles, the surface flow towards the SW is re-established and the formation of enriched levels of precursors (Figure 7) and ozone (Figure 8) in the 1000-2000 meters a.s.l. occurs again, in accordance with the regional recirculation processes reported in the literature for this area (Querol et al., 2018; Escudero et al.,

2019).

### 3.3.2. Advective pattern

As an example of an advective pattern, Figure 10 shows the plan view and the NE-SW cross section of $O_3$, NOx and VOCs levels during July 13th (Figure S9 shows the SE-NW cross section for day 13th and Figure S10 and Figure S11 represent the NE-SW cross section and the SE-NW cross section for day 20th, respectively in the Supplement). It can be seen that surface

$O_3$ levels at 3:00 UTC are around 8% lower than those of July 27th (accumulation), (average in the Madrid region of 39 ppb and 42 ppb, respectively) with maximum values along the Sierra de Guadarrama, where elevated terrain reaches layers rich in $O_3$ and precursors from the lower troposphere and from the residual layers formed the day before (Figure S11). This occurs

(also under accumulation conditions) when the PBL height is lower than the maximum height of the Sierra de Guadarrama. However, during advective periods, a stronger stratification of $O_3$ is observed during the early hours (3:00 – 9:00 UTC) due to the existence of more intense wind direction speed vertical gradients (relative to accumulation conditions), perfectly captured by the modeling system (Figure 11).

At 09:00 UTC, the local $O_3$ production downwind of the city is lower than during the accumulation periods (Figure S12), not only quantitatively but also in terms of the total area affected. This can be explained by the weather conditions (promoting dispersion) and the corresponding lower surface levels of the main precursors (5-8 ppb NOx and 15-20 ppb of VOCs on the day 13[th], compared to 10-15 ppb NOx and 30-40 ppb of VOCs during accumulation day 27[th]). At 15:00 UTC (Figure 10), the PBL height increases reaching 2,500-2,800 m altitude (compared to 4,000 m on day 27[th]). As the PBL grows, the vertical mixing dominates the wind-driven pollution displacement in the SW direction. Similarly to the dynamics described for accumulation conditions, this allows precursors and fresh $O_3$ to ascend and mix existing ozone in higher layers (Figure 10 and Figure 11). Nonetheless, the vertical mixing is lower during advective patterns, as observed in the ozone soundings (Figure 9), with the consequent difficulty of the boundary layer to incorporate $O_3$ from higher strata (beyond 4000 meters a.s.l.) in the central hours of the day. This results in lower $O_3$ mixing ratios at surface level under advective conditions, up to 60 ppb SW of Madrid City (Figure 10). As for the relative importance of local sources, Figure 11 shows that their contribution can reach nearly 30 ppb, similar to that under accumulation conditions. However, the area affected is clearly associated with the city plume and their contribution averaged over the region is smaller. In fact, our results point out that precursors advected can produce hourly peaks above 30 ppb outside the Madrid region.

### 3.4. Source apportionment assessment at the location of monitoring stations

A source apportionment assessment has also been carried out at the location of the air quality monitoring stations distributed throughout the simulation domain (Figure 1) to inform on the contributions of different sources in those points where air quality is routinely monitored. Differences are found depending on the type of station (urban, suburban and rural) and, consistently with the results discussed in the previous subsection, the type of circulation pattern (advective or accumulation). The results are summarized in Figure 12. As previously discussed, it shows the contribution of all anthropogenic emission sources (S01-03-04 to S08), biogenic emissions (SNAP 10-11) as well as boundary and initial conditions (BC, IC) and $O_3$ stratospheric transport (ST). Although 100% of emitting sectors have been tagged, Figure 12 shows as well the contribution from "others" (OTH). This contribution is typically negligible and relates to minor model interactions between sources and species not considered by the ISAM model. Details are fully explained in the documents provided with the model release (U.S. EPA, 2022).

Urban and suburban monitoring stations have a similar aggregated behavior. During the first hours of the morning, the initial and boundary conditions make up the totality of $O_3$ levels until 06:00 UTC approximately. After that time, $O_3$ generated from precursors emitted by local sources appears, reaching contributions up to 15 and 12 ppb for urban and suburban locations (28 and 22% of the total ozone, respectively) around 12:00 UTC. The road transport (14-10%) and the residential (2-4%) sectors are those with the highest contributions. The signal of anthropogenic sources is lower in rural monitoring stations. As an average, road traffic contributes a maximum of 5% (5 ppb), the residential sector 2% (2 ppb) and the use of solvents (VOC emissions) also around 2% in rural locations.

The results in Figure 12 demonstrate the persistent relevance of IC in all locations, but especially in rural locations. Even though the initial conditions contribute to $O_3$ levels throughout the day, the maximums values are found in the first hours (0:00 -5:00 UTC). As the day evolves, the influence of IC progressively decreases until they disappear at 21:00 UTC approximately. However, clear differences are found depending on the circulation pattern as illustrated for July 27[th] (accumulation) and July, 13[th] (advection). According to the model predictions, $O_3$ levels are greater during the accumulation period (and are reached slightly earlier), with maxima up to 68 ppb (17:00 UTC) in contrast with 52 ppb under advective conditions. Of note, the model reproduces observed $O_3$ temporal patterns quite consistently, but it misses the peak values during accumulation periods, as discussed in section 3.1.

It may be highlighted that the influence of residual layers of the previous day, tracked through the IC tag, is observed again at the central hours of the day is very significant under accumulation conditions (IC contribution of up to 12 ppb, around 18% of total $O_3$) while is practically missing for advective days. This relates to the enhancement of $O_3$ levels from reservoirs aloft discussed in section 3.3.1. that does not occur under advective conditions. Of note, and consistently with the analysis in section 3.3, we observe that the average contribution from local anthropogenic sources to $O_3$ peaks (around 16 hours local time) in urban locations in accumulation periods is higher than that of advective periods. That is true both, for absolute levels (18 ppb and 11 ppb, respectively) and relatively contributions (32% and 22%, respectively). These results point out that the source apportionment under unfavorable circulation patterns significantly differs from that for average or advective conditions and, consistently with previous studies (Lupaşcu et al., 2022: Qu et al., 2023) demonstrate that the influence of local sources is larger for high $O_3$ levels associated to stagnation conditions.

Nonetheless, clear differences are found for individual stations depending on their location relative to the city center and prevailing winds. In the supplement (Figure S14 to Figure S16) a stratification of the same results by station type and geographical quadrant (Figure S13) and distance to Madrid is shown. For instance, urban locations within Madrid municipality in the NE direction for the 27[th] July (accumulation) present much higher contributions from local sources that those of urban stations in the NW direction and further away from the metropolitan area (Figure S15). This variability suggests that the outcome of local measures may differ throughout the region and should be modeled under specific meteorological conditions and assesses specifically for each location of interest.

## 4. Conclusions

A high-resolution chemical-transport model has been used to investigate $O_3$ dynamics for a typical summer month (July 2016) in the Madrid Region. The model presents an acceptable performance and succeeds in reproducing the phenomena described in previous studies (Querol et al., 2018, Escudero et al., 2019), confirming that $O_3$ dynamics are conditioned by regional circulation patterns. Nonetheless, we found that model errors are larger for accumulation days and concentration peaks are underestimated. This may be related to an inadequate performance of the meteorological model under stagnation conditions. A novel implementation of CMAQ-ISAM (Shu et al., 2023) that attributes $O_3$ based reaction stoichiometry with all production and destruction reactions involved has been applied to perform a source apportionment of this non-linear, secondary pollutant under specific weather patterns. Our simulation shows that $O_3$ levels are dominated by non-local contributions (i.e., boundary conditions), representing around 70% of mean values across the region. Ozone reservoirs from previous days (label as initial conditions in our methodology) in the mid troposphere are also important to build up high $O_3$ levels in accumulation episodes,

representing the main difference with advective periods. The analysis, however, points out that precursors emitted by local sources play a more important role regarding the highest mixing ratios values, illustrated in this study by the 90$^{th}$ percentile. This suggests that the implementation of emission reduction strategies in the region may be more effective to control $O_3$ peaks than average values. This is particularly true under unfavorable, stagnation conditions associated with accumulation patterns when the highest $O_3$ values occur. According to our results, up to 35% of total $O_3$ may be originated from local sources, giving

a theoretical maximum reduction potential of 1-h values of approximately 25 ppb under these conditions. Among local sources, road traffic is the main contributor, accounting for 55% of local sources. Our results suggest that $NO_X$ emissions play a more important role than VOC emissions in the photochemical production of ozone. Nonetheless, we found that the use of solvents and other products, a significant source of VOCs emissions with high ozone formation potential, can explain up to 20% of the $O_3$ originated from local anthropogenic emissions in some locations. At the same time, our results point out that the contribution

of biogenic emissions is lower than that of anthropogenic sources (below 4% to total $O_3$ levels in this period), although they are responsible for 42.4% of total VOCs in the modeling domain. Emissions from other sectors play a minor role and $O_3$ transported from the Stratosphere within the model domain is negligible.

We also found significant variations in source apportionment patterns across station types and locations. This implies that high-resolution simulations under specific meteorological conditions should be performed to anticipate the potential outcome

on $O_3$ levels in different locations of the Madrid region.

Considering these results, future modeling efforts should be oriented to simulate the effect of specific measures both, local and in cooperation with other administrations to identify optimal emission abatement strategies. The modeling platform used in this study may be also helpful to assess sensitivities to different factors, including photochemical regimes or $NO_X$ and VOCs speciation for specific sources. Furthermore, the role of biogenic $NO_X$ and VOCs emissions may be further studied to

understand the implications for $O_3$ control strategies in the Madrid region.

**Code/Data availability**

The Community Multiscale Air Quality (CMAQ) and the Integrated Source Apportionment Method (CMAQ-ISAM) are an open-source development project of the U.S. EPA. The version used in this study (5.3.2) is freely available at: https://zenodo.org/records/4081737. Model outputs are available upon request to the authors.

**Author contributions**

RB and DP designed the research. DP and LT conducted the CMAQ modeling and data postprocessing. DP, JMA, LT, and RB analyzed the results. DP, RB and JMA wrote the paper with contributions from all authors. GS and SN provided support for the CMAQ model and reviewed the article.

**Competing interests**

The authors declare that they have no conflict of interest.

**Disclaimer**

The views expressed in this paper are those of the authors and do not necessarily represent the views or policies of the U.S. EPA.

**Financial support**

This study was carried out within AIRTEC-CM (urban air quality and climate change integral assessment) scientific program funded by the Directorate General for Universities and Research of the Greater Madrid Region (S2018/EMT-4329).

**Acknowledgements**

The authors gratefully acknowledge the Universidad Politécnica de Madrid (www.upm.es) for providing computing resources on Magerit Supercomputer.

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

**Table 1. Tagged sectors for the O$_3$ source apportionment analysis**

| Tagged sources | Description | Abbreviation |
|---|---|---|
| SNAP01 – SNAP03 – SNAP04 | Power generation (S01), Industrial combustion (S03) and Industrial processes without combustion (S04) | S01-03-04 |

| | | |
|---|---|---|
| SNAP02 | Non-industrial combustion plants | S02 |
| SNAP05 | Extraction and distribution of fossil fuels | S05 |
| SNAP06 | Use of solvents and other products | S06 |
| SNAP07 | Road Transport | S07 |
| SNAP08 | Other mobile sources and machinery | S08 |
| SNAP09 | Waste treatment and disposal | S09 |
| SNAP10 – SNAP11 | Agriculture and nature | S10-S11 |
| OTHER | Non-tagged emissions, including online computations (none in this study) | OTH |
| ICON | Initial conditions | IC |
| BCON | Boundary conditions | BC |
| PVO3 | Stratospheric ozone (potential vorticity) | ST |

**Table 2. Model performance statistics (dimensionless unless noted otherwise) by station type for ground-level O$_3$ concentration**

| Station type | FAC2 | MB ($\mu g \cdot m^{-3}$) | MGE ($\mu g \cdot m^{-3}$) | NMB | NMGE | RMSE ($\mu g \cdot m^{-3}$) | r | IOA |
|---|---|---|---|---|---|---|---|---|
| Industrial | 0.95 | 7.8 | 14.5 | 0.10 | 0.18 | 18.7 | 0.84 | 0.71 |
| Rural | 0.98 | -2.9 | 13.8 | -0.03 | 0.14 | 18.1 | 0.76 | 0.68 |
| Suburban | 0.94 | 2.4 | 17.15 | 0.03 | 0.20 | 23.3 | 0.74 | 0.69 |
| Urban background | 0.89 | 8.3 | 20.4 | 0.10 | 0.25 | 27.1 | 0.69 | 0.65 |
| Urban traffic | 0.88 | 10.8 | 19.9 | 0.14 | 0.25 | 26.5 | 0.73 | 0.65 |

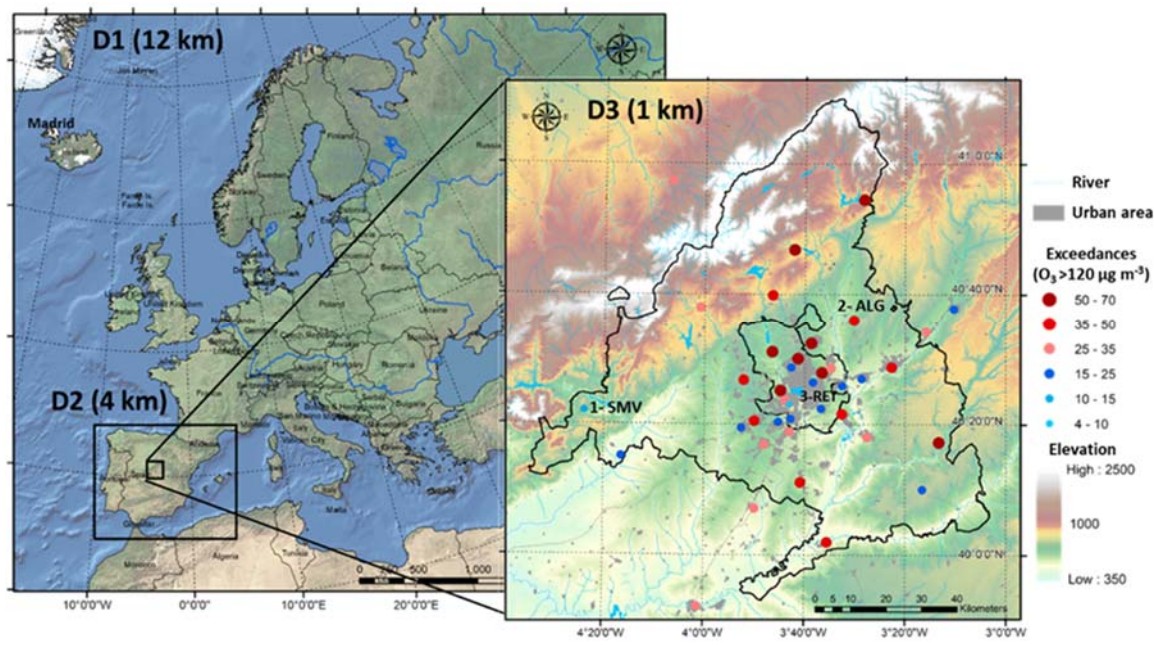

**Figure. 1. Modeling domains including the location of the air quality monitoring stations within the innermost domain and number of exceedances of the O₃ target value for protection of human health (MDA8 > 120 µg·m⁻³) in 2016.**

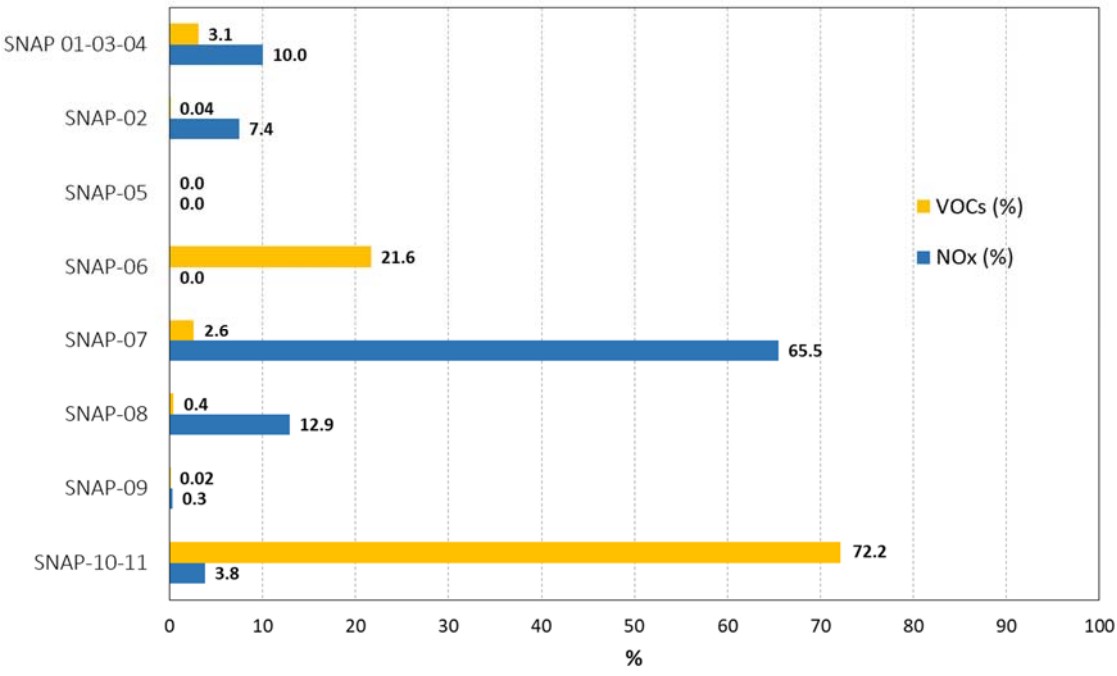

**Figure. 2. NOₓ and VOC emissions of tagged sectors for July 2016 (percentage over total emissions in the modeling domain) for the source apportionment analysis.**

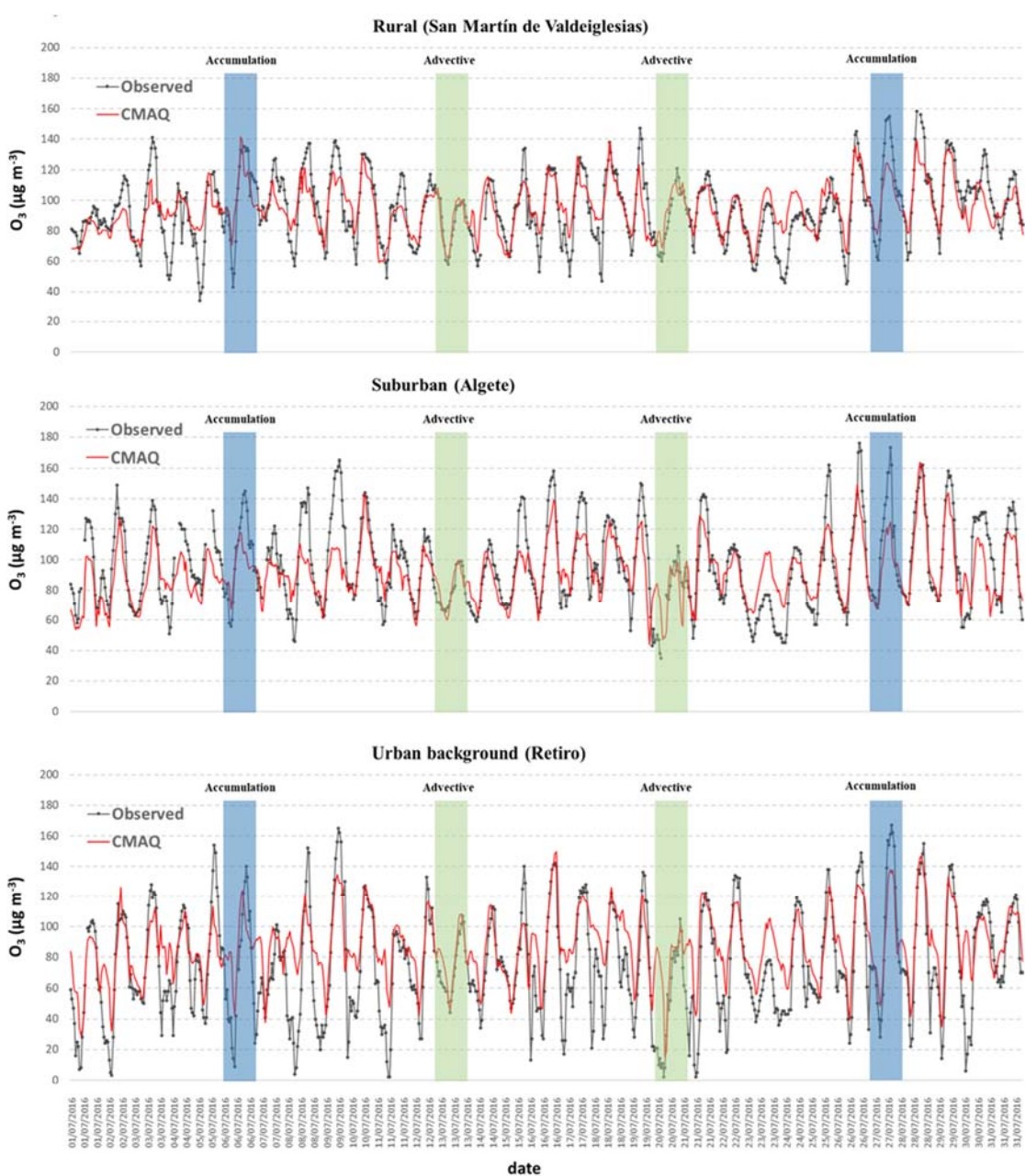

**Figure. 3. Observed and predicted concentration series for selected locations (1-SMV: a rural location in the southwestern area of Madrid region, 2-ALG: a suburban location in the northeastern area of Madrid region and 3-RET: an urban background site in Madrid city center).**

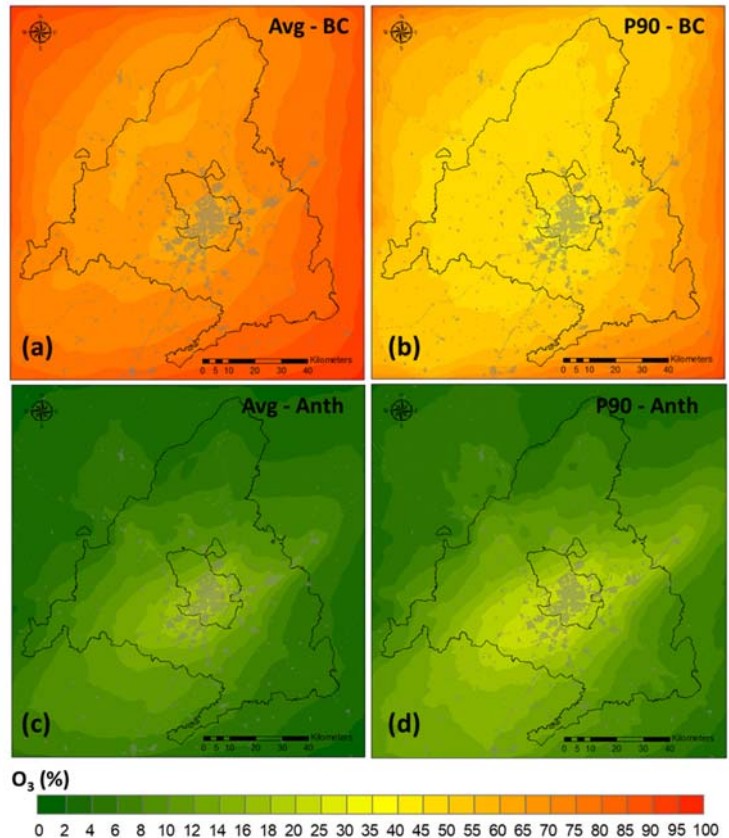

O₃ (%)

0 2 4 6 8 10 12 14 16 18 20 25 30 35 40 45 50 55 60 65 70 75 80 85 90 95 100

**Figure 4. Contribution (%) of BC to O₃ concentration: (a) monthly average and (b) 90th percentile. Contribution (%) of local anthropogenic emissions to (c) monthly average and (d) 90th percentile.**

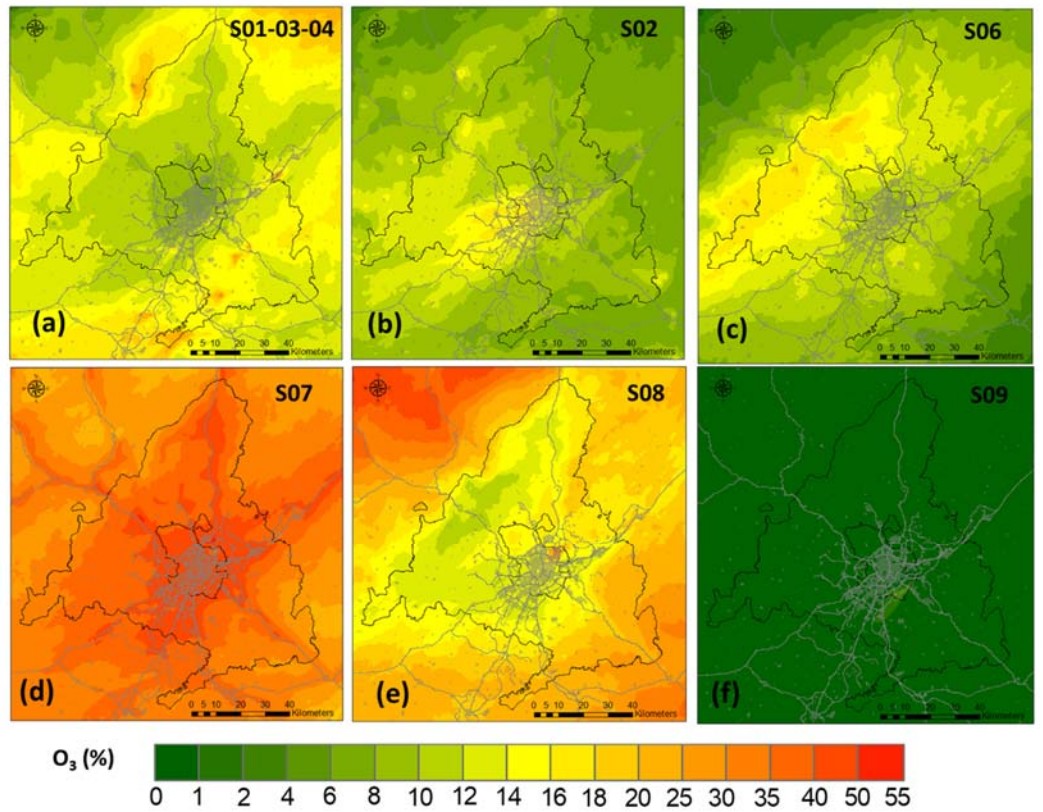

O₃ (%)

0 1 2 4 6 8 10 12 14 16 18 20 25 30 35 40 50 55

**Figure 5. Percentage contribution to the 1-hour 90th O₃ percentile of the main emitting sectors with respect to the total anthropogenic contribution.**

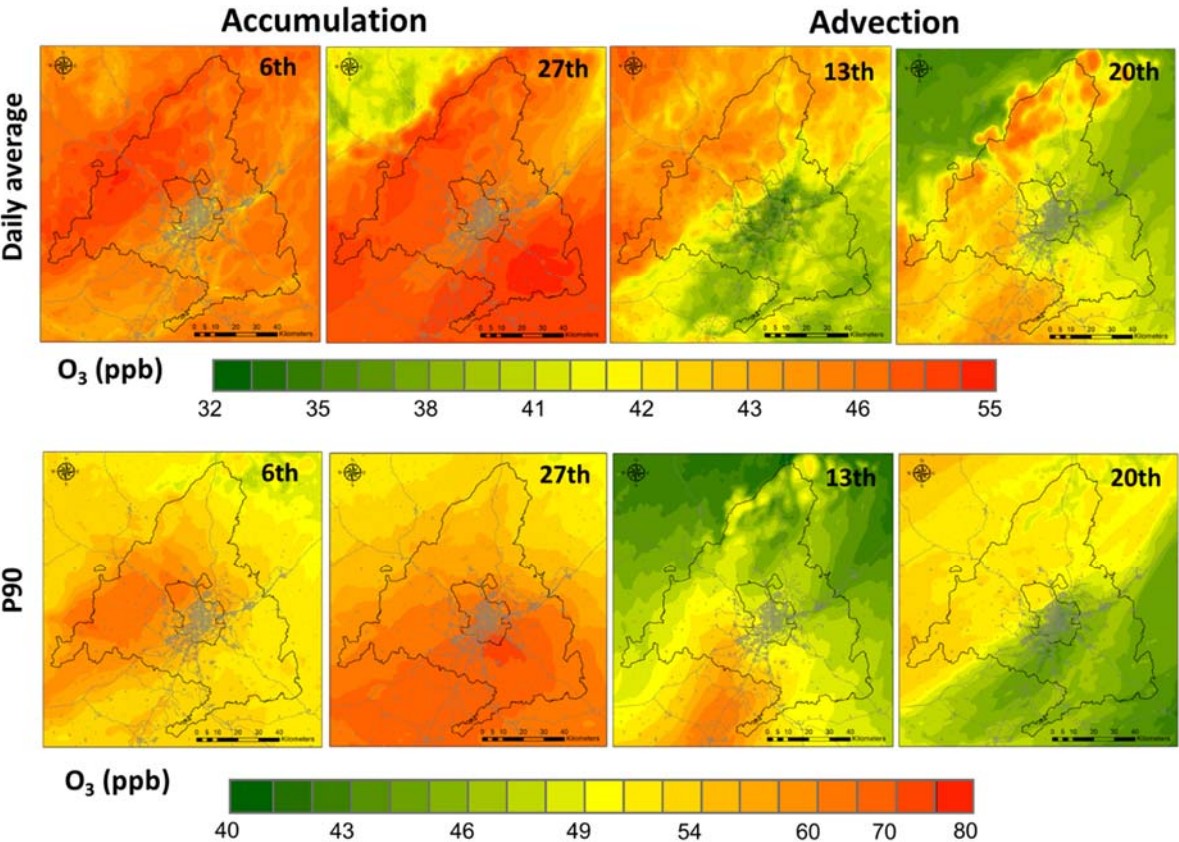

 **Figure 6. Daily mean (top) and 90th percentile (bottom) of O₃ levels (ppb) during accumulation (6th and 27th July, 2016) and advective (13th and 20th July, 2016) periods**

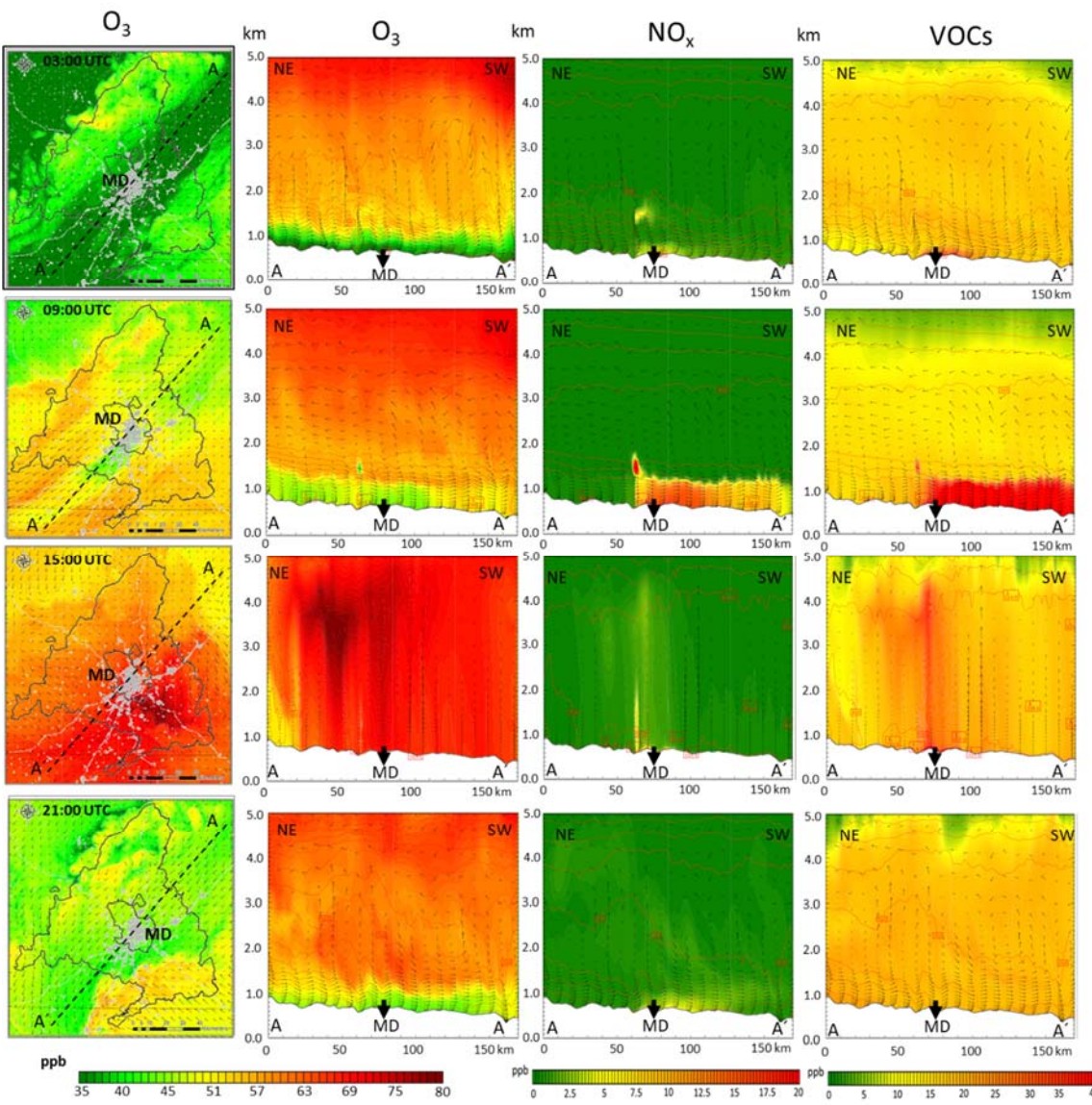

**Figure 7. Accumulation period: evolution during July 27th, 2016. From left to right, plan view and NE-SW cross section (up to 5 km height) O₃ mixing ratios (ppb), NOx (ppb) and VOCs (ppb) at the 3:00, 9:00; 15:00, 21: 00 UTC hours. MD = Madrid City.**

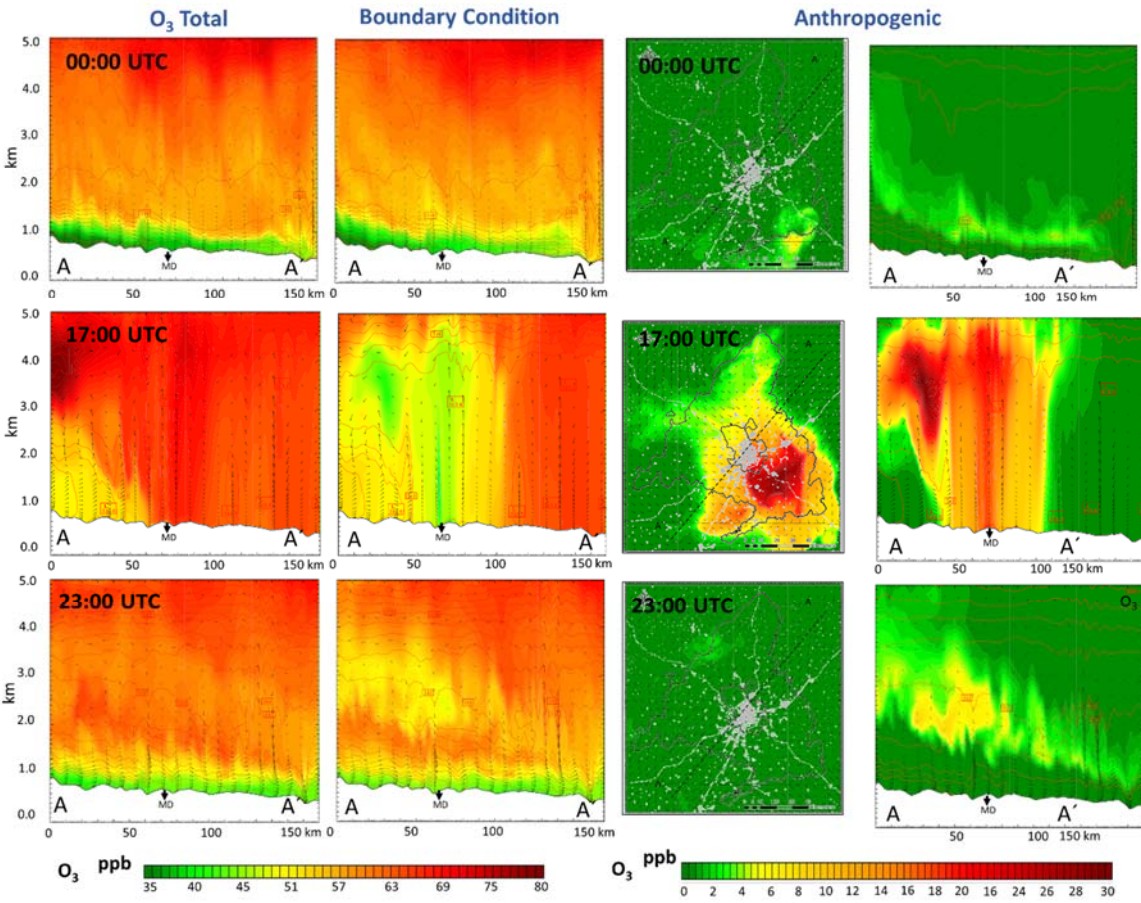

**Figure 8.** Hourly O₃ mixing ratios profiles (at 0:00, 17:00; 23:00 UTC) for the NE-SW cross section and contribution of BC and anthropogenic local emissions on July 27th, 2016 (accumulation). MD = Madrid City.

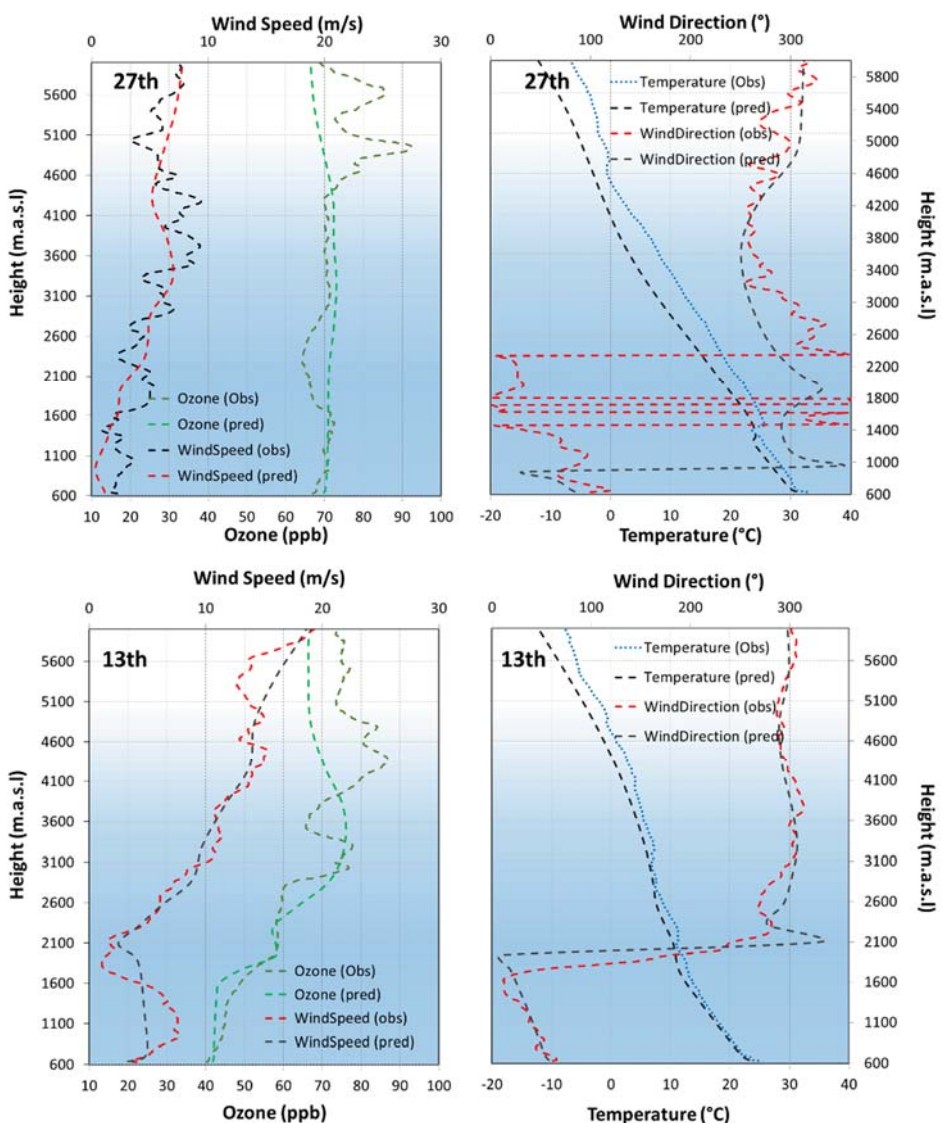

**Figure 9. Vertical profiles (noon UTC) of O₃ mixing ratios, temperature, wind speed and wind direction for July 27th (accumulation, up) and the July13th (advective, down).**

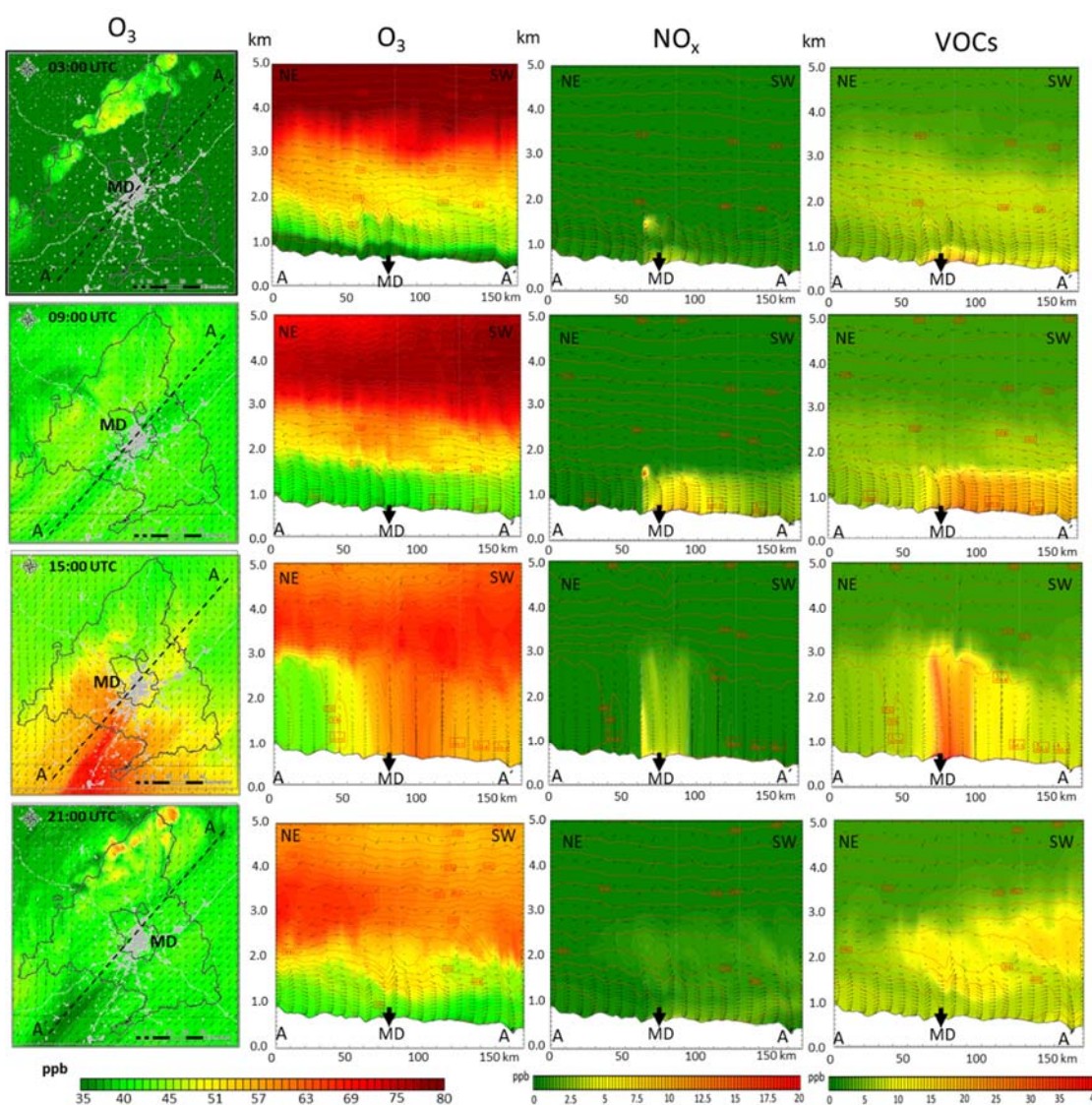

**Figure 10. Advective period: hourly evolution during July 13th, 2016. From left to right, plan view and NE-SW cross section (up to 5 km height) O₃ mixing ratios (ppb), NOx (ppb) and VOCs (ppb) at the 3:00, 9:00; 15:00, 21: 00 UTC hours. MD = Madrid City.**

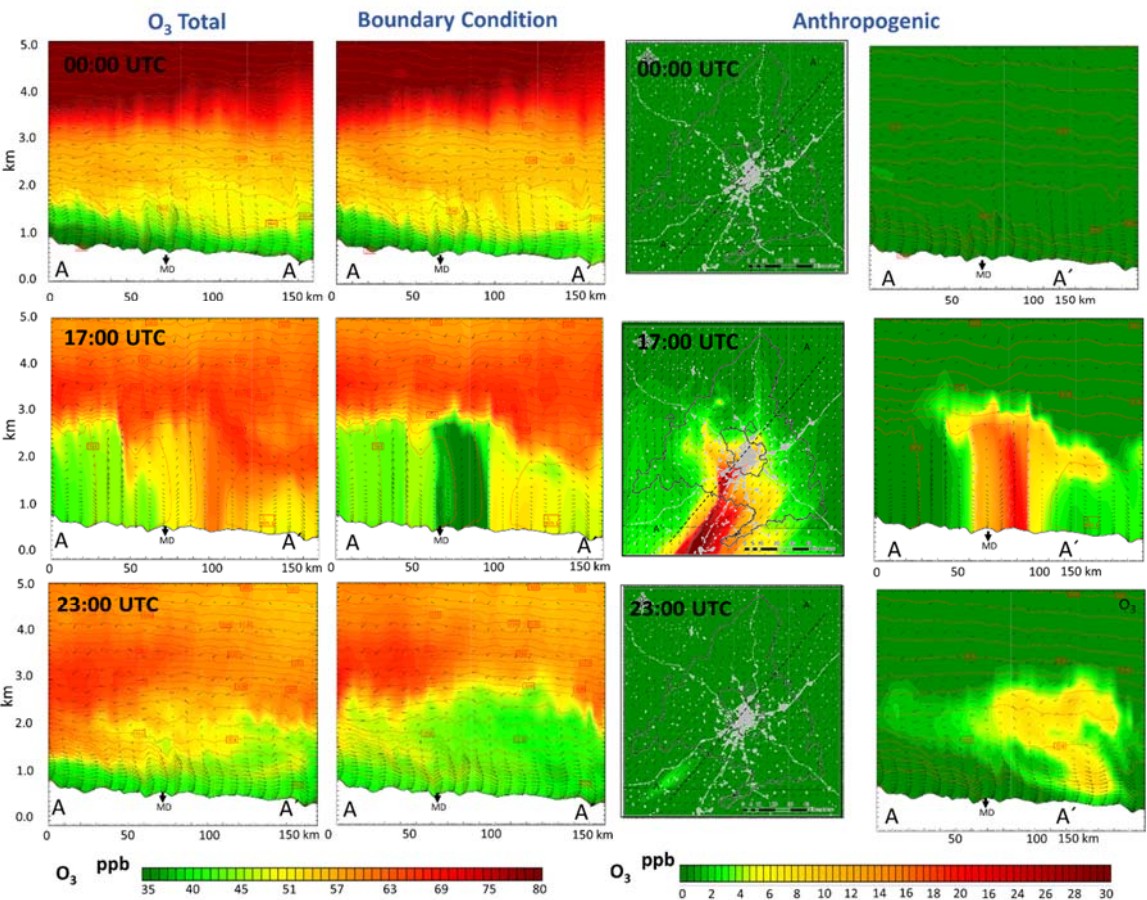

**Figure 11. Hourly O₃ mixing ratios (at 0:00, 17:00; 23:00 UTC) for the NE-SW cross section and contribution of BC and anthropogenic local emissions on July 13th, 2016 (advection). MD = Madrid City.**

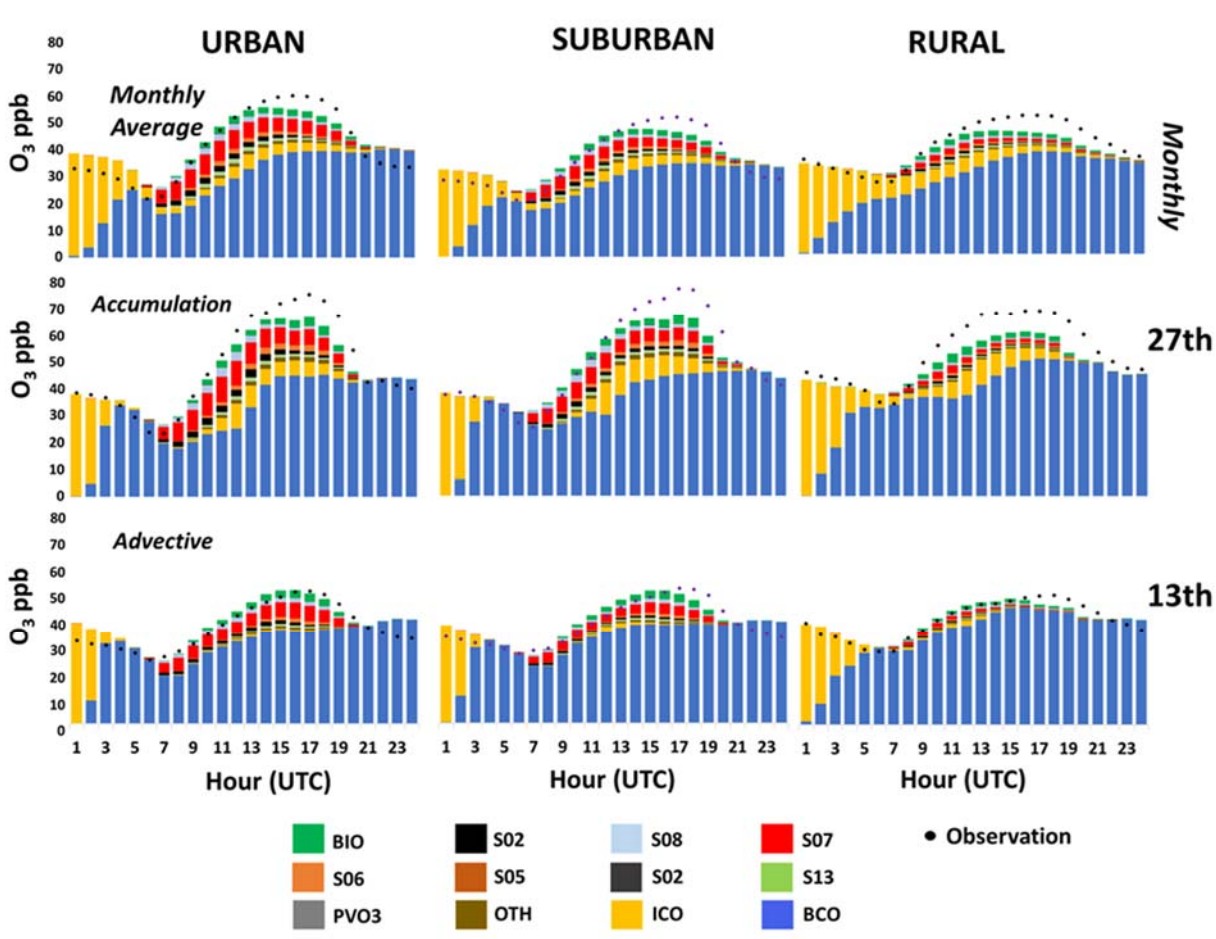

Figure 12. Hourly contribution for the monthly average (top) and specifically for accumulation (27th July, 2016) and advective (13th July, 2016) days (middle and bottom, respectively).