# Peer review of "Summertime tropospheric ozone source apportionment study in the Madrid region (Spain)"

_EGUsphere, 2023_

## Referee Comment (RC2)

**Review article: Summertime tropospheric ozone source apportionment study in Madrid (Spain)**

The paper describes a modeling study that investigates the source of summertime surface ozone ($O_3$) in Madrid, Spain, using an integrated source apportionment method within the Community Multiscale Air Quality model (CMAQv5.3.2). The paper highlights the importance of local sources (road traffic) in the build-up of $O_3$ during peak events, which tends to happen during anticyclonic stagnation conditions in summertime. Suggesting that local measurements aim to reduce $O_3$ precursors could have a positive effect during such periods. In general, the findings are valuable for the understanding of the $O_3$ build-up mechanism in the region and, consequently, for policy decision-making.

The narrative in the abstract suggests that the study focused on the source contributions to urban $O_3$ pollution. However, the results include the contribution to $O_3$ in different chemical environments (urban, suburban, and rural); therefore, I believe that the scope needs to be clearly stated in the document. Also, I feel that the source apportionment method is loosely defined in the methods session. Expanding the description of the method with an example of the mechanics would strengthen the paper. It also feels that the discussion of the model evaluation has been skipped, and something is missing in the paper to convince the reader of the model's capabilities to reproduce the meteorology and chemical environment of the period study. It would be helpful to show a time series of $O_3$ at some representative sites (urban, regional, and suburban) to show the diurnal and day-to-day variability (e.g., hourly ozone) and the model performance. This is important as it gives the reader a general idea of the $O_3$ evolution and the pollution episodes, which are investigated throughout the study. Once these issues have been addressed, along with the points below, the manuscript will be suitable for publication.

**General Comments**

When you say that the contribution from biogenic emissions is relatively small and therefore excluded from the analysis, are you referring to the contribution to regional scale O3? Could you consider including an evaluation of biogenic VOCs, e.g., a time series comparison of isoprene or a statement regarding the performance of the MEGAN model? Biogenic VOCs, in particular isoprene, are important contributors to O3 formation during photochemical O3 episodes, particularly in rural areas but also in urban environments (Dunker et al., 2016), due to their reactivity and abundance. You concluded that biogenic sources are responsible for 42.4% of the total VOCs domain-wide, so an important impact from this source should be expected.

Evaluation of the model is reported in the supplemental material and loosely mentioned in the manuscript. Please adjust section 3 and provide a quantitative statement of the model performance for meteorology and chemistry (especially O3) for the model domains, along with

some plots. This could be a spatial contour plot showing the model and observed mean or P95 of O3 or time series of O3 at some representative sites.

**Specific Comments**

Line 32:33: 'These measures, however, have failed to significantly improve ozone (O3) ambient concentration levels'. I feel this is a strong affirmation that needs to be re-phrased, taking into account the nonlinearity nature of O3 formation and the different aspects related to the concentrations observed in different chemical environments (e.g., urban, rural, and suburban) as well as the effects of emissions reductions such as the urban decrement.

Line 140: It would be helpful to have a paragraph describing how experiments were designed, for instance, how the chemical cycling is performed and how often the meteorology is restarted.

Line 249: The link provided does not work

Line 360: The link provided does not work

Dunker et al., 2016.https://doi.org/10.1016/j.atmosenv.2016.09.048

---

## Author Comment (AC1)

**Final author replay to the editor**

Dear editor, thank you for the opportunity to revise our submission and provide responses to the points raised by both anonymous reviewers. We acknowledge them for fruitful critics and for their suggestions that helped us to improve our work.

In the document attached we provide point-to-point responses to each of their questions and comments. The author comments are structured according to ACP guidelines and follow the recommended sequence: comments from the referees (RC1 and RC2) are shown in blue, then we provide our responses in black and the main changes made in the manuscript are included in red. We respond to both reviewers in the same document. Although we provide a common response to the points raised in some cases we duplicate the corresponding explanations to facilitate the work of both reviewers.

Since we made substantial changes throughout the document, we include the revised version of our submission in full (manuscript and supplementary material with changes highlighted in red) at the end of this document for the sake of clarity.
* * *
**RC1**

The manuscript "summertime tropospheric ozone source apportionment study in Madrid (Spain) by de la Paz et al. presents a ozone source apportionment study using ISAM in the CMAQ model over the Madrid region for July 2016.

While the topic of the paper is very interesting and fits into the scope of ACP, the manuscript lacks many important information and the results are discussed in an insufficient way. Therefore, I recommend to reject the manuscript. Please find below a more detailed review given some major and minor comments the authors should consider before the plan to resubmit the publication.

Thank you for the time devoted to our manuscript and for pointing out potential ways to improve our work and specific issues that need further discussion or clarification.

Reading the paper I had several major concerns which needs to be clarified.

1) The authors apply a new version of ISAM in CMAQ 5.3.2 which (to my knowledge) has not been used in similar publications before. According to the authors this new version attributes ozone to all involved precursors and NOT to the limiting precursor (e.g. NOx or VOC). Sadly, the manuscript is lacking any details on the new method. I guess the method is somehow similar to the method presented by Grewe et al., 2017? The results presented in the manuscript heavily depend on this method. Therefore, the method either needs to be presented before in a scientific publication (e.g. an update of Kwok et al., 2015 and not a youtube video) which is cited by the authors or the manuscripts needs to include a detailed presentation/discussion of the revised method.

To our knowledge this is, in fact, one of the early applications of this version of CMAQ-ISAM and constitutes one of the novelties of the paper. However, there is a now publication, which became available during the review process of our work, that fully details the latest implementation of CMAQ-ISAM, provides a sample application, and compares results to other apportionment techniques. Additionally, the manuscript was modified to provide some brief explanation of CMAQ-ISAM, and more importantly to direct the reader to Shu et., (2023) for a more detailed view of the model. This replaces the reference to the "youtube video", which is a recording of an invited presentation at the CMAS Annual Conference (a highly regarded scientific event among CMAQ users worldwide) that accompanies the conference proceedings. So, Napelenok et al. (2020b) has been replaced by the reference to this paper published in Geoscientific Model Development. Consequently, Napelenok et al. (2020a) is referred as Napelenok et al. (2020) in our revised manuscript (line 136). Considering the comments from the reviewers, we included a brief discussion of source apportionment techniques and recent applications in the introductory section (lines 75-87) and focus on the description of CMAQ ISAM specifically in section 2.1 (lines 135 to 153 of the revised manuscript).

The paragraph included in section 1 reads:

[revised manuscript text omitted]

2) Given the new method, I am missing a detailed discussion of the method and the results in comparison to previous publications (see for example Butler et al., 2018 for a detailed discussion of

many ozone tagging methods). Moreover, I am missing a critical discussion of the model results. As example, Fig.4 shows a contributions of more than 14 % of SNAP6 (solvents) to ozone over the mountain range north of Madrid. Given my understanding of the method and results of similar methods I wonder about this high contribution. How can this be explained? Moreover, I wonder about the small contribution of biogenic emissions, even though they account for a large fraction of the VOC emissions. If the method attributed ozone to all precursors, they should account for a larger fraction (?) Please clarify.

As pointed out in the previous response, we further discuss the CMAQ-ISAM version used in our study and include a reference for a more detailed view of the model (Shu et al., 2023). To provide a better context of this study, we included a more elaborated discussion of source apportionment techniques and recent applications in the introduction.

As for the critical discussion of our results, we have revised our paper to provide a better insight and explain our results. To clarify the two specific points raised by Reviewer #1 in this comment, we elaborate and discuss our results in section 3.2 (3.1 in the original submission) about the contribution of the main VOC sources, both anthropogenic and biogenic. Regarding the first, we would like to clarify that Figure 4 (Figure 5 in the revised manuscript) shows that emissions from the SNAP 6 sector (solvents) may contribute more than 14% to $O_3$ P90 (close to 20% in some points), but this refers to the anthropogenic contribution, so it is in the range of 2% of total $O_3$ P90 levels. From the scientific literature and the speciation profiles used for VOC emissions in this sector, we propose that this contribution relates to the high reactivity of individual species within these emissions such as aromatics, that have a large ozone formation potential (OFP). In our discussion we cite other studies that point in that direction too. As for the spatial distribution of this contribution, it reaches the maximum impact (in absolute values) 20-30 km away from Madrid City center in the southwest direction (see Figure AC1, included in the supplementary material as Figure S2 in our revised submission). Of note, that does not correspond exactly with the mountain range north of Madrid, but the southside mountain foothills (Figure 1). However, the maximum relative contribution is found in the northwest direction (Figure 5 in the revised manuscript) because the total contribution of anthropogenic emissions -dominated by $NO_X$- is smaller in that area. This is related to the slowest chemistry of VOCs in comparison with that of $NO_X$ and similar results have been found for other urban environments such as New York City.

[Figure]

**Figure AC1. Absolute contribution (ppb) to the monthly mean 1-hour 90$^{th}$ O$_3$ percentile of the SNAP 06 sector (use of solvents and other products) and biogenic emissions.**

This is discussed in the passage between lines 325 and 340 within section 3.2 of our revised manuscript:

"… of the Adolfo Suárez Madrid-Barajas airport. This suggests that NO$_X$ emissions play a more important role than VOC emissions in the photochemical production of ozone, in concurrence with previous source apportionment studies (Dunker et al., 2016; Butler et al., 2018; de la Paz et al., 2020). Nonetheless, the importance of controlling anthropogenic VOC emissions to prevent high O$_3$ episodes has been noted in previous studies (Cao et al., 2022), even in regions with strong biogenic emissions (Coggon et al., 2021). In addition to the contribution of BVOC previously discussed, anthropogenic VOC had also an influence on O$_3$ levels during July 2016 in the Madrid region. While the spatially-averaged attribution of O$_3$ to SNAP 06 is only 1.5 ppb with maximum contributions of 3 ppb at specific locations (southwest of Madrid as shown in Figure S2 and Figure S5), emissions from the use of solvents and other products can reach values up to 20% of total anthropogenic contributions to O$_3$ P90 (Figure 4c). This is comparable to the contribution of all industrial sources combined (SNAP01-03-04). This may be related to the high OFP of aromatics within SNAP 06 VOC (Meng et al., 2022) and is consistent with the findings of Oliveira et al., (2024) that attributed 64% or total OFP to the solvent sector (relative to that of total anthropogenic VOC) in densely urbanized areas such as Madrid. Coggon et al., (2021) also found that consumer and industrial products

(included in SNAP 06 group) are important precursors of ozone in urban areas, were typically present a VOC-sensitive regime. Nonetheless, they found that $O_3$ formation may take a few hours and the maximum contributions of VOC emitted in New York City to occur a few tens of km away, close to $NO_X$-limited areas. Our high-resolution analysis indicates that a similar process may take place in Madrid too. The rest of the sectors analyzed (SNAP05 and SNAP09) have negligible contributions (around 0.05 ppb as an average over the Madrid region)."

We also reflect on our results regarding the contribution of biogenic emissions and discuss them in the context of previous studies. We reformulated the narrative not to underestimate the influence of biogenic VOCs (BVOC) and compare our results with the findings of previous studies in the literature. We make clear that the direct comparison is not possible since the interpretation depends on the specific source apportionment methodology used and the specific model domain and scale of application. As shown in Figure AC1 (=Figure S2), the maximum contribution to $O_3$ of biogenic emissions occurs in the central area of the Madrid region, even though vegetation emissions in that more densely populated area are smaller. That is consistent with previous studies that reported stronger contributions of BVOC to $O_3$ levels in VOC-limited areas. On the contrary, the production of $O_3$ away from anthropogenic high-intensity areas is limited due to the unavailability of $NO_X$.

As for the relative importance in comparative terms with other sources, we found that global and continental studies usually attribute a more important role to BVOC in the explanation of $O_3$ budgets.  Nonetheless, our results seem consistent for other studies in the Iberian Peninsula or the Madrid region specifically. Nonetheless, we highlight the need to use caution comparing results from different studies, so the reader is not misled by methodological differences. We connect this discussion with the reactivity of specific VOCs for a more consistent view of the role of VOCs, focusing on isoprene -see response to first general from Reviewer #2, when we compare CMAQ predictions with observed values for this specie (see Figure AC2, included in the supplementary material of our revised submission as Figure S3)- as one of the key BVOC in the atmospheric photochemistry. We believe this discussion helps to clarify why the contribution of BVOC is smaller than that of all anthropogenic sources combined but larger than that of the SNAP 06 alone.

[Figure]

**Figure AC2. Comparison of isoprene ground-level mixing ratios predicted by CMAQ (left) and measurements made in Majadahonda (suburban site) by Querol et al., (2018) (right). Both graphs present the hourly values during the day averaged over the period July 5th and July 19th. The source of the right-hand panel is Pérez et al., (2016).**

Reference (added to the supplementary material):

- Pérez, N., A. Alastuey, C. Reche, M. Ealo, G. Titos, A. Ripoll, M.C. Minguillón, F. J. Gómez-Moreno, E. Alonso-Blanco, E. Coz, E. Díaz, B. Artíñano, S. García dos Santos, R. Fernández-Patier, A. Saiz-López, F. Serranía, M. Anguas-Ballesteros, B. TemimeRoussel, N. Marchand, D. C. S. Beddows, R. M. Harrison y X. Querol. Campaña intensiva de medidas de UFP, O3 y sus precursores en el área de Madrid: medidas en superficie., https://www.miteco.gob.es/content/dam/miteco/es/calidad-y-evaluacion-ambiental/temas/atmosfera-y-calidad-del-aire/anexo_informea33_madrid_tcm30-561368.pdf (last access: [January 22, 2004]), 2016.

The discussion about biogenics is included at the beginning of the revised section 3.2 and it reads as it follows:

[revised manuscript text omitted]

In addition, the manuscript is lacking an overview of the definition of the different source attribution sectors (e.g. as table etc). From Fig 11 it seems that 12 different source sectors are considered.

Tagged sectors, i.e. emission sources $O_3$ is attributed to, are presented in Figure 2. The discussion in lines 158-166 included in the original manuscript has been slightly revised to clarify the sectors tagged as well as other $O_3$ sources (not related to emitting activities) discussed in Figure 11 (Figure 12 in the revised manuscript). That passage now it reads:

"The share of $NO_X$ and VOCs emissions of each SNAP (Selected Nomenclature for Air Pollution) groups is summarized in Figure 2. Emissions from power generation and industrial activities (SNAP 01, SNAP 03 and SNAP 04) were merged due to their limited presence in this modeling domain (and noted as S13 in Figure 12). Since emissions from agriculture (SNAP 10) in the region are only significant for VOCs from plants, they have been tagged along biogenic VOC (BVOC) emissions from vegetation (SNAP 11) (and labeled as BIO in Figure 12). Consequently, 8 emission sources were tagged for the source apportionment analysis of ambient $O_3$ in the region, as reflected in Figure 2. They account for the totality of emissions in the modeling domain although the main precursors originate from road traffic (SNAP 07) and solvent use (SNAP 06), with a total share of 65% $NO_X$ and 49% VOCs, respectively. While emissions from the residential, commercial and institutional sector (SNAP 02) account for nearly 19% of annual $NO_X$ emissions, they are produced almost exclusively in winter and are therefore, negligible in summer.

In addition to the attribution of $O_3$ ambient levels to the emissions within the modeling domain, hereinafter referred to as local sources, the contribution of boundary conditions (BC) and initial conditions (IC) are also estimated in this study (labeled as BCO and ICO in Figure 12). Considering the typical $O_3$ daily patterns and the variability of circulation patterns, the latter refer to the initial mixing ratios on a daily (24 hour) basis, i.e., each day is run separately using the outputs from the previous day as IC."

Besides making an explicit reference to Figure 12 (Figure 11 in the original submission) for the sake of clarity, we added the following paragraph when Figure 12 is introduced (Lines 423-427 or the revised manuscript):

"The results are summarized in Figure 12. As previously discussed, it shows the contribution of all anthropogenic emission sources (S13 to S08), biogenic emissions (BIO) as well as boundary and initial conditions (BCO, ICO) and $O_3$ stratospheric transport (PVO3). Although 100% of emitting sectors have been tagged, Figure 12 shows as well the contribution from "others" (OTH). This relates to second-order interactions between sources (U.S. EPA, 2022). This represents a negligible fraction in this study, i.e. ISAM could attribute the virtual totality of $O_3$ to any of the other sources."

Reference added:

- U.S. EPA: Community Multiscale Air Quality (CMAQ) model v5.4 User Guide, Office of Research and Development, U.S. EPA, https://github.com/USEPA/CMAQ/tree/5.4/DOCS/Users_Guide (last access: [January 22, 2004]), 2022.

3) I am missing any new results. The large importance of boundary conditions to ozone levels over the Iberian Peninsula have been reported by e.g. Pay et al., 2019. Also the larger importance of regional emissions to high ozone values have been presented in previous publications (maybe not focusing on Madrid). I like the detailed investigation of source attribution results for specific weather patterns, however, for a scientific publication in ACP more detailed analyses are needed in my opinion and the author need to highlight new findings in more detail.

We believe that this paper contributes to improve current knowledge about the attribution of air pollution to emission sources in general and adds considerable value to understand ozone pollution dynamics in the Madrid region in particular. We have revised our manuscript to highlight the main novelties of our work. They are as follows:

- Pay et al. (2019) presented a countrywide source apportionment analysis for a typical high-$O_3$ summer period. While they used the same chemical-transport model (CMAQ), there are relevant differences (see summary in Table AC1) that allow us to further delve into the specific $O_3$ dynamics for the Madrid region. Both studies agree on the dominant role of BC and consistently identify road traffic as the main local contributor to the production of $O_3$, especially regarding peak levels. However, we provide an estimate of the contributions of other sources not explicitly considered in previous studies, including solvent use, BVOC emissions or stratospheric transport. More importantly, our approach allows us to identify different apportionment structure depending on local circulation conditions and demonstrates the influence of $O_3$ generated in the previous 24 hours under stable atmospheric conditions (accumulation pattern). In addition, we report significant differences across our modelling domain depending on these meteorological patterns as well. Our study is by no means redundant of previous ones. On the contrary, it makes perfect sense considering the recommendation of Pay et al., (2019) or Escudero et al., (2019) regarding the need for detailed quantification of contributions to high $O_3$ concentrations considering the influence of local sources and topographical and meteorological conditions to effectively inform local strategies as well as exploring the apportionment for different phenomenology of high-$O_3$ episodes. In addition, we think our

methodology may be illustrative for other regions worldwide to perform local source apportionment studies that may support subsequent $O_3$ plans.

- This is one of the first applications of this CMAQ-ISAM implementation (Shu et al., 2023) and may support future source apportionment studies elsewhere using this tool
- Our study quantifies for the first time the implications of local circulation patters identified in previous studies (Querol et al., 2018, Escudero et al., 2019) for $O_3$ source apportionment, providing useful information for the future development of plans and strategies, specially for short-term action plans
- For instance, we provide a first estimate of the theoretical maximum reduction of maximum $O_3$ ambient levels (approximately 25 ppb for 1-hour maximum) under unfavorable conditions, a result of large significance for the design of local strategies to reduce $O_3$ levels
- Considering these new findings, we propose some options for future research that may further improve our understanding of complex $O_3$ dynamics and effectively inform new policies

Table AC1. Comparison of the methodology of this study with that of Pay et al. (2019)

| Feature | Pay et al. (2019) | Our study |
|---|---|---|
| Domain and resolution | Spain, 4 km x 4 km | Madrid region, 1 km x 1 km |
| Temporal domain | 10 days (21 - 31 July 2012) | 1 month (July 2016), 24-hours runs to identify the role of IC Specific analyses for relevant weather patterns |
| SA method | CMAQ-ISAM based on sensitivity regime (Kwok et al., 2015) - option 5 in current release of CMAQ (v 5.4)- | CMAQ-ISAM based on equal assignment for all reactants (Shu et al., 2023) -option 1 in current release of CMAQ (v 5.4)- |
| Tagged sectors | 5 (power generation, industry, road transport, off-road mobile sources, others) + IC + BC | 8 (power generation and industry, non-industrial combustion plants - residential, commercial and institutional emissions-, road transport, off-road mobile sources, waste treatment, agriculture and nature emissions) + IC + BC + stratospheric transport |
| Model performance assessment | Aggregated | Aggregated and individual (42 monitoring stations) |
| Analysis at specific monitoring-sites/locations | 2 | 18 |

We have revised our paper to provide a more comprehensive analysis of the results (see changes made related to previous questions) and tried to highlight the contribution of our work. This is specifically reflected in the revised version of the conclusions, copied below:

"A high-resolution chemical-transport model has been used to investigate $O_3$ dynamics for a typical summer month (July 2016) in the Madrid Region. The model presents an acceptable performance and succeeds in reproducing the phenomena described in previous studies (Querol et al., 2018, Escudero et al., 2019), confirming that $O_3$ dynamics are conditioned by regional circulation patterns. Nonetheless, we found that model errors are larger for accumulation days and concentration peaks are underestimated. This may be related to an inadequate performance of the meteorological model under stagnation conditions. A novel implementation of CMAQ-ISAM (Shu et al., 2023) that attributes $O_3$ based reaction stoichiometry with all production and destruction reactions involved has been applied to perform a source apportionment of this non-linear, secondary pollutant under specific weather patterns. Our simulation shows that $O_3$ levels are dominated by non-local contributions (i.e., boundary conditions), representing around 70% of mean values across the region. Ozone reservoirs from previous days (label as initial conditions in our methodology) in the mid troposphere are also important to build up high $O_3$ levels in accumulation episodes, representing the main difference with advective periods. The analysis, however, points out that precursors emitted by local sources play a more important role regarding the highest mixing ratios values, illustrated in this study by the 90th percentile. This suggests that the implementation of emission reduction strategies in the region may be more effective to control $O_3$  peaks than average values. This is particularly true under unfavorable, stagnation conditions associated with accumulation patterns when the highest $O_3$ values occur. According to our results, up to 35% of total $O_3$ may be originated from local sources, giving a theoretical maximum reduction potential of 1-h values of approximately 25 ppb under these conditions. Among local sources, road traffic is the main contributor, accounting for 55% of local sources. Our results suggest that $NO_X$ emissions play a more important role than VOC emissions in the photochemical production of ozone. Nonetheless, we found that the use of solvents and other products, a significant source of VOCs emissions with high ozone formation potential, can explain up to 20% of the $O_3$ originated from local anthropogenic emissions in some locations. At the same time, our results suggest that the contribution of biogenic emissions is lower than that of anthropogenic sources (below 4% to total $O_3$ levels in this period), although they are responsible for 42.4% of total VOCs in the modeling domain. Emissions from other sectors play a minor role and $O_3$ transported from the Stratosphere within the model domain is negligible.

We also found significant variations in source apportionment patterns across station types and  locations. This implies that high-resolution simulations under specific meteorological conditions should be performed to anticipate the potential outcome on $O_3$ levels in different locations of the Madrid region.

Considering these results, future modeling efforts should be oriented to simulate the effect of specific measures both, local and in cooperation with other administrations, to identify optimal emission abatement strategies. The modeling platform used in this study may be also helpful to assess sensitivities to different factors, including photochemical regimes or $NO_X$ and VOCs speciation for specific sources. "

4) The authors mix the physical quantities "concentration" and "mixing ratio". They use the term concentration and use the unit ppb which suggest a (volume) mixing ratio. Please clarify the used physical quantity. Similarly, Fig. 2 does not give any physical quantities for the emissions. In addition,

please clarify what emissions of NOx and VOC are. Are they given in amount of N, NO, NO2 and C or NMHC?

It is usual in the scientific literature and even in the air quality regulation to use the term "concentration" to refer to near-ground relative abundance of a given pollutant both, when volume/volume (ppb) or $\mu g/m^3$ are used. Nonetheless, we agree that mixing ratio is the correct term, especially when discussing vertical profiles. Following the reviewer's suggestion, we refer to "concentration" only when discussing the results of our model with observed values (expressed in $\mu g/m^3$, standardized at a temperature of 293 K and an atmospheric pressure of 101,3 kPa) to reduce ambiguity. Consequently, we have revised the caption of original Figures 5, 6, 7, 8, 9 and 10 (as well as Figures in the supplementary material: S1, S2, S3, S4, S5, S6, S7, S8 and S9). Please, see the revised version of our manuscript included after the response to reviewers.

As for Figure 2, we revised the corresponding caption to clarify that shows the contribution as a percentage over total emissions in the region for each pollutant. The purpose of that figure is to illustrate the emission share and the relative importance of each source in our inventory and not present emissions in absolute terms. Just for clarification, the emission inventories used in this research are compiled according to the EMEP/EEA methodology: EMEP/EEA air pollutant emission inventory guidebook 2019. Technical guidance to prepare national emission inventories (https://www.eea.europa.eu/publications/emep-eea-guidebook-2019), which is a standard in Europe. In addition to the references to the specific inventories used, we included this one in the revised manuscript to let the reader understand the conceptual basis of the emission estimates (line 192). According to this methodology, NOx emissions account for both NO and $NO_2$ emissions collectively expressed as $NO_2$ mass. According to the technical definition of NMVOC emissions for inventory reporting, they "comprise all organic compounds except methane which at 273.15 K show a vapor pressure of at least 0.01 kPa or which show a comparable volatility under the given application conditions" (AQEG, 2020) and intends to represent the total mass of organic compounds that are capable of producing photochemical oxidants by reaction with nitrogen oxides in the presence of sunlight.

Both, the caption and the legend in Figure 2 have been updated too:

[Figure]

| Snap-01 | Combustion in the production and transformation of energy | Snap-07 | Road Transport |
| Snap-02 | Non-industrial combustion plants | Snap-08 | Other mobile sources and machinery |
| Snap-03 | Industrial combustion plants | Snap-09 | Waste treatment and disposal |
| Snap-04 | Industrial processes without combustion | Snap-10 | Agriculture |
| Snap-05 | Extraction and distribution of fossil fuels and geothermal energy | Snap-11 | Other sources and sinks (nature) |
| Snap-06 | Use of solvents and other products | | |

**Figure. 2. NO$_X$ and VOCs emissions of tagged sectors (percentage on an annual basis) for the source apportionment analysis.**

5) Parts of the manuscript are confusing and missing a proper proof-reading. As example, on P5l197 the authors write that Fig. 4 shows "average contributions". The description of Fig.4, however, indicates that contribution to the 90$^{th}$ percentile of ozone are given. Moreover, I find it very confusing, that the authors only show contributions to ozone attributed to anthropogenic origin. I suggest to always show contributions with respect to total ozone. Otherwise results are very hard to compare to other studies and readers might be confused.

We can confirm that the caption of Figure 4 (Figure 5 in the revised version of our manuscript) is correct. The reference to Figure 4 (page 5, line 197) read in the original submission "Figure 4 shows

the apportionment of each emission sector for local sources. Road transport (SNAP07) is the most influential sector, with an average contribution in the Madrid region of 41% and with maximum contributions of around 55%, located in the proximity of the main communication routes (Figure 4d)." The discussion here refers to high $O_3$ values, illustrated by the $90^{th}$ percentile. We slightly changed the paragraph to avoid confusions like the one pointed out by reviewer #1. Now it reads:

"Figure 5 shows the apportionment to P90 of each emission sector for local sources. Road transport (SNAP07) is the most influential sector, contributing 41% to P90 as an average over the Madrid region. The contribution of this sector (relative to local sources) reaches values up to 55% in the proximity of the main communication routes (Figure 5d).

We appreciate the second suggestion made by Reviewer #1, but we think the current approach may be more effective to communicate our findings. Since contribution from sources outside the modeling domain dominate $O_3$ levels, the apportionment of specific sectors is presented as a percentage of the total contribution of local sources. At the same time, we provide information on the absolute contribution, in terms of ppb. We think this may make the interpretation easier and it may be more informative to support the design of strategies. We believe that the clarifications made regarding this issue, as well as the interpretation of initial conditions (see response to comment #7) and an extended discussion of source apportionment methods (see response to comment #1) have help clarifying our methodology and results.

The paper has been proofread again trying to make the discussion more accessible considering the example highlighted by Reviewer #1. We also corrected several typos and mistakes in the text (see revised version after RC2).

6) In the last subsections the authors present a comparison with measurements. This comparison shows an underestimation of ozone simulated by the model under accumulation conditions during 13 -19 UTC, however the authors do not discuss this model bias. How does it affect the source apportionment results? It seems that the model underestimated local ozone production under this stagnant conditions. To my opinion, the manuscript should start with a model evaluation and discuss the source attribution results critically with respect to the model performance.

Following the suggestions from both reviewers, we included a new section (3.1) to provide a better view of the model performance assessment. We keep the detailed results of model assessment for each air quality monitoring station in the supplementary material (Table S3 in the revised version of our submission) because we think it helps the interpretation of site-specific results and complements the information given by aggregated statistics (Table 1). All statistics have been revised and harmonized. In addition to this, we added two new tables (Table S4 and Table S5) to illustrate the differences on model performance (both CMAQ and WRF) depending on the circulation pattern. Following the suggestion of Reviewer #2 we also show the comparison of observed and modeled $O_3$ series for 3 representative sites (pinpointed now in the revised version of Figure 1) as Figure 3. Besides illustrating the capabilities of the model and the reason for the statistical results obtained, it serves to present the features of the study period. Although a detail investigation of the causes for model discrepancies with observation is out of the scope of this contribution, we think it helps understanding potential reasons for performance differences found.

As discussed in this new section, the difficulty of the meteorological model to reproduce wind fields under very weak forcing conditions (accumulation patterns) may contribute to the larger bias found in CMAQ outputs for that circulation type. We acknowledge this limitation and put our results in context with a critical discussion of our results and those from other relevant studies. We think this new section demonstrates a reasonable performance to study ground-level $O_3$. Furthermore, the results shown in Figure 8 (Figure 9 now) suggest a robust model performance also to describe $O_3$ mixing rations aloft. We think this is enough to build confidence in the ability of the system to accurately describe ozone typical features and thus, we believe the modeling tool is fit for the purpose of the research at hand.

The new section 3.1 is as follows:

**"3.1 Ozone levels during the study period and model evaluation**

While this period was hotter and dryer than most of recent summers, July 2016 may be representative of typical summer conditions in the Madrid region and included a concatenation of characteristic local circulation patterns (Plaza et al., 1997) with direct implications on ground-level $O_3$ (Querol et al., 2018; Escudero et al., 2019). Figure 3 presents both observed and modeled concentration series at representative points (Figure 1), and shows the venting and accumulation days identified in Querol et al., (2018). The time series demonstrate that $O_3$ levels are significantly lower under venting conditions, although significant differences are found depending on the location, which supports the need to use high-resolution modeling systems to analyze pollution dynamics in the Madrid region. On the other hand, accumulation patterns tend to produce higher concentrations (up to 175 µg/m$^3$), especially during July 27$^{th}$.

It can be observed that the model is able to reproduce the temporal patterns, as confirmed by the high correlation coefficients (r) and index of agreement (IOA) shown in Table 1. The statistical evaluation demonstrates a reasonable model performance, yielding better statistical results than recent simulation studies in this domain. Pay et al. (2019) reported an aggregated correlation coefficient of 0.66 and mean bias (MB) of 22.5 ug/m$^3$ for the central region of the Iberian Peninsula. In this study, we obtained an average r value of 0.74 and a MB of 6.2 ug/m$^3$. Of note, 95.2% and 66.7% of the r values for the locations of the 42 monitoring stations used in this study are larger than 0.6 and 0.7, respectively while the overall normalized mean bias (NMB) is only 9%. The results for a series of common statistics (Borge et al., 2010) for each of the monitoring sites in our modeling domain can be found in Table S3. The model, however, may have some difficulties capturing the amplitude of observed $O_3$ series and fails to accurately reproduce concentration peaks on some days. This is evidenced by the relatively large error in comparison with the bias (23% and 9%, respectively as an average over the 42 monitoring stations in the modeling domain). In the supplementary material (Table S4), we present a separate model performance assessment for accumulation and advective patterns showing that the main differences among them relate to errors, both MGE and RMSE that are systematically higher for accumulation periods. This may be related to the limitations of the meteorological model to depict atmospheric circulation during stagnant conditions suggested by Pay et al., (2019). Even when WRF was found to outperform other

models for this particular episode (Escuedero et al., 2019), the ability to reproduce wind direction and wind speed clearly deteriorates for accumulation periods, as shown in Table S5,

As expected, results are poorer for urban background and traffic locations, since the typical spatio-temporal representativeness of the measurements in such locations is not comparable with that of a mesoscale modeling system, even with 1 km$^2$ spatial resolution."

[Figure]

**Figure. 3. Observed and predicted concentration series for selected locations (1-SMV: a rural location in the southwestern area of Madrid region, 2-ALG: a suburban location in the northeastern area of Madrid region and 3-RET: an urban background site in Madrid city center).**

And the new tables included in the supplementary material are:

Table S4. Model performance statistics (dimensionless unless noted otherwise) by station type and circulation pattern for ground-level O₃ concentration.

| Station | Pattern | n | FAC2 | MB ($\mu gm^{-3}$) | MGE ($\mu gm^{-3}$) | NMB | NMGE | RMSE ($\mu gm^{-3}$) | r | IOA |
|---|---|---|---|---|---|---|---|---|---|---|
| Rural | Accumulation | 240 | 0.98 | -6.7 | 15.29 | -0.06 | 0.14 | 18.83 | 0.83 | 0.66 |
| | Advective | 232 | 0.98 | 3.1 | 9.31 | 0.04 | 0.11 | 12.97 | 0.83 | 0.73 |
| | Other | 3211 | 0.98 | -3.0 | 14.01 | -0.03 | 0.15 | 18.30 | 0.75 | 0.67 |
| Suburban | Accumulation | 474 | 0.96 | -4.8 | 20.24 | -0.05 | 0.20 | 26.69 | 0.76 | 0.68 |
| | Advective | 468 | 0.92 | 7.3 | 13.59 | 0.10 | 0.19 | 19.69 | 0.75 | 0.68 |
| | Other | 6412 | 0.94 | 2.6 | 17.18 | 0.03 | 0.20 | 23.22 | 0.73 | 0.68 |
| Urban background | Accumulation | 669 | 0.89 | 2.4 | 23.46 | 0.03 | 0.26 | 31.04 | 0.69 | 0.66 |
| | Advective | 670 | 0.89 | 11.4 | 16.95 | 0.17 | 0.25 | 22.34 | 0.72 | 0.60 |
| | Other | 9014 | 0.89 | 8.5 | 20.41 | 0.11 | 0.25 | 27.08 | 0.68 | 0.65 |
| Industrial | Accumulation | 96 | 0.95 | 4.7 | 16.40 | 0.05 | 0.18 | 20.15 | 0.86 | 0.73 |
| | Advective | 96 | 0.97 | 9.1 | 12.55 | 0.13 | 0.18 | 15.26 | 0.82 | 0.65 |
| | Other | 1278 | 0.95 | 7.9 | 14.54 | 0.10 | 0.18 | 18.79 | 0.83 | 0.71 |
| Urban traffic | Accumulation | 510 | 0.91 | 3.5 | 20.09 | 0.04 | 0.22 | 25.81 | 0.79 | 0.69 |
| | Advective | 522 | 0.87 | 15.8 | 18.22 | 0.25 | 0.28 | 24.55 | 0.69 | 0.55 |
| | Other | 7086 | 0.87 | 11.0 | 19.98 | 0.14 | 0.25 | 26.72 | 0.73 | 0.65 |

Table S5. Model (WRF) performance statistics by circulation pattern for basic meteorological variables

| Variable | Pattern | FAC2 | MB | MGE | NMB | NMGE | r | IOA |
|---|---|---|---|---|---|---|---|---|
| Temperature (T2) | Accumulation | 1.00 | -1.4 K | 2.0 K | -0.05 | 0.07 | 0.92 | 0.81 |
| | Advection | 1.00 | -0.5 K | 1.5 K | -0.02 | 0.06 | 0.96 | 0.86 |
| | Other | 1.00 | -0.8 K | 1.6 K | -0.03 | 0.06 | 0.96 | 0.85 |
| Wind speed (WS10) | Accumulation | 0.63 | 0.9 m/s | 1.7 m/s | 0.31 | 0.63 | 0.30 | 0.33 |
| | Advection | 0.78 | 0.7 m/s | 1.5 m/s | 0.17 | 0.37 | 0.59 | 0.55 |
| | Other | 0.71 | 0.5 m/s | 1.3 m/s | 0.18 | 0.46 | 0.58 | 0.55 |
| Wind direction | Accumulation | 0.61 | -34.3 ° | 90.7 ° | -0.24 | 0.63 | 0.26 | 0.55 |
| | Advection | 0.87 | 6.5 ° | 34.5 ° | 0.05 | 0.25 | 0.79 | 0.81 |
| | Other | 0.77 | -9.2 ° | 60.8 ° | -0.06 | 0.38 | 0.53 | 0.68 |

7) Given the importance of emissions from the previous day for ozone formation I wonder why the authors attribute them to "IC". Wouldn't it be better to account them also sectorwise?

That is the approach followed by other source apportionment studies, but we think our methodology serves us better considering the temporal span of the period analyzed (a whole month), the typical diurnal cycle of O₃ and the goal of characterizing this attribution under specific meteorological conditions. This is another novelty of our methodology that may be better suited to provide useful information for decision making, especially for the design of short-term action plans intended to control ozone peaks. This is an important point and we added an explicit discussion at the end of section 2 (lines 209-217) to make it clear before discussing the results:

"In addition to the attribution of $O_3$ ambient levels to the emissions within the modeling domain, hereinafter referred to as local sources, the contribution of boundary conditions (BC) and initial conditions (IC) are also estimated in this study (labeled as BCO and ICO in Figure 12). Considering the typical $O_3$ daily patterns and the variability of circulation patterns, the latter refer to the initial mixing ratios on a daily (24 hour) basis, i.e., each day is run separately using the outputs from the previous day as IC. This is a difference with most previous source apportionment studies that analyze shorter periods (Pay et al., 2019) or specific high concentration events (Lupaşcu et al., 2022; Zhang et al., 2022). While this may hinder the comparability of our results, this methodological option may be appropriate considering the temporal span of the period analyzed (a whole month), the typical diurnal cycle of $O_3$ and the goal of characterizing this attribution under specific meteorological conditions. This helps understanding differences on $O_3$ source apportionment depending on regional circulation patterns (Zhang et al., 2023) and explicitly considering the influence of vertical transport of $O_3$ from residual layers form previous days that may lead to rapid increases of $O_3$ concentrations near the surface (Qu et al., 2023 and references within). Therefore, this approach may be better suited to provide useful information for decision making, especially for the design of short-term action plans intended to control ozone peaks."

References added:

- Lupaşcu, A., Otero, N., Minkos, A., and Butler, T.: Attribution of surface ozone to NOx and volatile organic compound sources during two different high ozone events, Atmos. Chem. Phys., 22, 11675–11699, https://doi.org/10.5194/acp-22-11675-2022, 2022.
- Qu, K., Wang, X., Cai, X., Yan, Y., Jin, X., Vrekoussis, M., Kanakidou, M., Brasseur, G. P., Shen, J., Xiao, T., Zeng, L., and Zhang, Y.: Rethinking the role of transport and photochemistry in regional ozone pollution: insights from ozone concentration and mass budgets, Atmos. Chem. Phys., 23, 7653–7671, https://doi.org/10.5194/acp-23-7653-2023, 2023.
- Zhang, S., Zhang, Z., Li, Y., Du, X., Qu, L., Tang, W., Xu, J., and Meng, F.: Formation processes and source contributions of ground-level ozone in urban and suburban Beijing using the WRF-CMAQ modelling system, Journal of Environmental Sciences, 127, 753-766, https://doi.org/10.1016/j.jes.2022.06.016, 2023.
- Zhang, Y., Yu, S., Chen, X., Li, Z., Li, M., Song, Z., Liu, W., Li, P., Zhang, X., Lichtfouse, E., and Rosenfeld, D.: Local production, downward and regional transport aggravated surface ozone pollution during the historical orange-alert large-scale ozone episode in eastern China, Environ Chem Lett, 20, 1577–1588, https://doi.org/10.1007/s10311-022-01421-0, 2022.

Nonetheless, to provide additional information on the question raised by Reviewer #1, we looked at the source apportionment structure at the end of each day (used as initial condition for the following) to understand what the contribution within the "ICO" label is sectorwise. As an average, the virtual totality of $O_3$ at midnight comes from BC (as already shown is Figures 11, S11, S12 and S13 in our original submission). The trace of IC from local sources after 24 hours is very weak, specially under advective conditions, when the influence of the ozone from the day before is negligible (Figure AC3).

[Figure]

**Figure AC3. Contribution of anthropogenic sources at the end of the day (23:59 PM) for advective conditions (July 13th and July 20th) and accumulation conditions (July 6th and July 27th)**

However, we identified the moments and locations when the contribution of the remaining sources was higher at the end of the day and looked into the apportionment structure. The results, shown in Figure AC4 reveal that the breakdown of IC is very similar to that discussed throughout the paper for the mean and P90 levels. Therefore, we conclude that the approach follow for IC was key to track the relative importance of $O_3$ from previous days, but it does not distort the sectoral analysis.

[Figure]

**Figure AC4. Sectoral breakdown of the contribution of IC (BC excluded)**

8) Some of the reference seems to be not adequate. As an example, P3l118 cites Borgee et al., 2022 (https://doi.org/10.1016/j.scitotenv.2013.07.093), but I can't find "tagging" nor "ISAM" in the whole paper. Maybe I misunderstood something, but the authors should check the manuscript carefully.

We carefully checked all the references in the manuscript and made minor changes, conveniently tracked in the revised version. In addition to a significant number of new references, we added missing cites in the original submission such as Butler et al. (2020) https://doi.org/10.5194/acp-20-10707-2020 to the references list. However, the particular reference pointed out by Reviewer #1 seems to be correct. https://doi.org/10.1016/j.scitotenv.2013.07.093 is the DOI of Borge et al. (2014) that presents the first source apportionment study made in Madrid using a sensitivity approach (brute force). However, Borge et al. (2022) refers to a comparison of the single-perturbation method (or brute force) with two implementations of CMAQ-ISAM; that of version 5.0.2 and the one used here, corresponding to version 5.3.2, that is completely pertinent for the discussion at hand. Thanks to this comment we realized the year was missing in the reference, something we corrected in the revised version of our manuscript. We also realized that https://doi.org/10.1016/j.atmosenv.2022.119258 was mistakenly included within the references and was removed since it is not adequate here.

Correct reference for Borge et al. (2022):

- Borge, R., de la Paz, D., Cordero J.M., Sarwar, G., Napelenok, S.: Comparison of Source Apportionment Methods to attribute summer tropospheric O3 and NO2 levels in Madrid (Spain) 21st International Conference on Harmonisation within Atmospheric Dispersion Modelling for Regulatory Purposes.  HARMO21, Aveiro, Portugal, 27-30 September 33-37, 2022.

Reference removed:

-

Some minor comments:

- Introduction: I am missing a discussion of similar source attribution studies (globally, for Europe) and a discussion of comparable source attribution methods.

We included a brief discussion on source apportionment techniques and recent applications in the introductory section (lines 75-87) (see response to Question #1). In addition, we believe that other additions to the description of our methodology the results previously discussed would help the reader to frame our work and better understand our findings.

- Fig 2: COVs instead of VOC

The legend in Figure 2 has been corrected (see response to Question #4 -second part-)

-p5l183 I wonder why the contribution of biogenic emissions is so small (see also major comments above).

We include a more detailed analysis of the contribution of biogenic emissions that would definitively contribute to better understand the role of biogenic emissions in this study and potential differences with previous works (see response to Question #2 -last part-)

- P3l109 What is the temporal resolution of the boundary conditions?

We clarify that the temporal resolution of boundary conditions from hemispheric CMAQ is 1 hour (line123 of the revised manuscript)

"…the mother domain receives 1 hour-resolution, dynamic chemical boundary conditions from hemispheric CMAQ (Mathur et al., 2017) simulations. "

- p2l45 You mean STE is projected to increase? Please clarify.

We don't mean that STE is projected to increase in this particular location. According to the global scale simulations of Meul et al., (2018) and Banerjee et al., (2016) (among others) downward transport form $O_3$ from the Stratosphere to the Troposphere is expected to increase significantly due to dynamic and chemical changes in the atmosphere induced by climate change. We made this clarification (lines 46-47) in the revised version of our manuscript:

"…expected to increase in the future globally (Meul et al., 2018; Banerjee et al., 2016) due to dynamic and chemical changes in the atmosphere induced by climate change."

To our knowledge our work presents the first explicit apportionment of stratospheric $O_3$ locally transported to ground level in the region. As illustrated in Figure 11 (and analogous ones in the supplement) of our original submission, the average contribution of stratospheric $O_3$ (PVO3) is negligible. As shown in Figure AC5 (added in the supplementary material as Figure S4), the maximum 1-hour contribution in the region is less than 0.4 ppb. It should be bear in mind that PVO3 accounts only for stratospheric ozone downward fluxes within the modeling domain, and presumably a significant part of the contribution of BC relates to STE. We have included this discussion in section 3.2 (lines 297-305):

"In addition, we tagged stratospheric ozone (PVO3 in Figure 12) due to the influence of vertical injections on ground level $O_3$ levels (Hsu et al., 2005) and the potential contribution reported in this region for specific extraordinary ozone levels (San José et al., 2005). Pay et al. (2019) hypothesize that stratosphere-troposphere exchange (STE) may have played a significant role towards the end of July 2016 in the Iberian Peninsula. According to our results, however, the direct transport of $O_3$ from the stratosphere in our modeling domain was negligible in this period, with 1-hour maximum contributions below 0.4 ppb in the southwest end of the Madrid region (see Figure S4). This contrasts with remarkably higher contributions reported in other areas of Europe (Lupaşcu et al., 2022) and those from global simulations for similar latitudes (Butler et al., 2018). It should be noted that here we account for $O_3$ STE exclusively within our innermost nested domain and part of the $O_3$ attributed to BC may be related to contributions from the Stratosphere in other regions. "

References added:

- Butler, T., Lupascu, A., Coates, J., and Zhu, S.: TOAST 1.0: Tropospheric Ozone Attribution of Sources with Tagging for CESM 1.2.2, Geosci. Model Dev., 11, 2825–2840, https://doi.org/10.5194/gmd-11-2825-2018, 2018.
- Lupaşcu, A., Otero, N., Minkos, A., and Butler, T.: Attribution of surface ozone to NOx and volatile organic compound sources during two different high ozone events, Atmos. Chem. Phys., 22, 11675–11699, https://doi.org/10.5194/acp-22-11675-2022, 2022.
- San José, R., Stohl, A., Karatzas, K., Bohler, T., James, P., and Pérez, J.L.: A modelling study of an extraordinary night time ozone episode over Madrid domain, Environmental Modelling & Software, 20(5), 587-593, https://doi.org/10.1016/j.envsoft.2004.03.009, 2005.

[Figure]

**Figure AC5. Maximum 1-hour attribution of stratospheric transport (PVO3) to ground level**

- P2l47ff I am missing a discussion of the role of the non linearity of the ozone chemistry which lead to an increase of the ozone production efficiency when emissions are reduced. The authors should consider to add this point including a discussion of the relevant literature.

The ozone production efficiency is indeed affected by changes on emissions that induce changes on the oxidation capacity of the atmosphere (Jung et al., 2022; Jung et al., 2023). However, we think that the point raised by Reviewer #1 is particularly relevant for source apportionment studies based on sensitivity approaches (Dunker et al., 2016; Sartelet et al., 2022). Therefore, addressing this question explicitly may be out of the scope of our paper, that presents a diagnosis study based on emission tagging. Nonetheless, we implicitly address this issue in the revised version of our manuscript when we discuss the contribution of VOC emissions, both from SNAP 06 and biogenic sources (please, see response to Question #2 and related additions to the text).

[Figure]

**NO emissions (tn)**
- 0.0000135 - 0.0000225
- 0.0000106 - 0.0000134
- 0.0000087 - 0.0000105
- 0.0000073 - 0.0000086
- 0.0000061 - 0.0000072
- 0.0000050 - 0.0000060
- 0.0000041 - 0.0000049
- 0.0000033 - 0.0000040
- 0.0000025 - 0.0000032
- 0.0000017 - 0.0000024
- 0.0000008 - 0.0000016
- 0.0000000 - 0.0000007

**Figure AC6. Soil NO emissions (t) estimated in our modeling domain for July 2016**

- p5l185 But Pay et al, 2019 applies the "old" ISAM tagging, right? So I would expect a difference with the new approach? Please discuss.

That is correct (see Table AC1 within the response to Question #3). Nonetheless, the latest version of CMAQ keeps both options (see response to Question #1). We include further discussion of the differences of the different methods. The resulting source attribution may differ in a different degree depending on the pollutant, the scale of analysis and the specific features of each modeling domain. In this particular case, Borge et al., (2022) addresses the comparison of both approaches (along with the single-perturbation method) (Figure AC7). They found significant differences on the attribution of $NO_2$ but not for $O_3$. Both methods identified road traffic (SNAP 07) as the major local anthropogenic contributor but CMAQ-ISAM attributes a larger share of ground-level $O_3$ to this source when the apportionment is based on an equal assignment for all reactants. This version attributes larger contributions to local sources in general (vs BC) (Figure AC8), specially around Madrid metropolitan area. That may be one of the reasons why, for instance, the attribution to biogenic sources in Pay et al., (2019) is apparently smaller than the one obtained in our current study.

[Figure]

**Figure AC7. Comparison of road traffic (SNAP 07) contribution to monthly average O₃ levels depending on the source apportionment methodology. Source: Borge et al., (2022)**

[Figure]

OA – anthropogenic emissions other than road traffic

**Figure AC8. Comparison average O₃ levels source apportionment in two different locations of the Madrid region (city center -left- and remote rural location -right-) depending on the source apportionment methodology. Source: Borge et al., (2022)**

**RC2**

Review article: Summertime tropospheric ozone source apportionment study in Madrid (Spain)
The paper describes a modeling study that investigates the source of summertime surface ozone (O3) in Madrid, Spain, using an integrated source apportionment method within the Community Multiscale Air Quality model (CMAQv5.3.2). The paper highlights the importance of local sources (road traffic) in the build-up of O3 during peak events, which tends to happen during anticyclonic stagnation conditions in summertime. Suggesting that local measurements aim to reduce O3 precursors could have a positive effect during such periods. In general, the findings are valuable for the understanding of the O3 build-up mechanism in the region and, consequently, for policy decision-making.

Thank you for the time devoted to our manuscript, your reassuring remarks and the suggestions to improve our work.

The narrative in the abstract suggests that the study focused on the source contributions to urban O3 pollution. However, the results include the contribution to O3 in different chemical environments (urban, suburban, and rural); therefore, I believe that the scope needs to be clearly stated in the document.

Air pollution is a multi-scale problem and urban air quality is affected by emissions from different geographical areas. At the same time, pollutants released in cities have an impact beyond the urban areas. In the case of Madrid, pollution dynamics is strongly determined by emissions from the metropolitan area (Borge et al., 2014) that affects the whole region. That is particularly true when dealing with secondary pollutants such as ozone. While we think this is reflected in the abstract, we agree that the title of our paper may be misleading and, therefore it has been slightly changed to "Summertime tropospheric ozone source apportionment study in the Madrid region (Spain)" in the revised version of our manuscript.

We revised the introductory section to clearly define the scope of our research and provide a better context of previous source apportionment studies. The changes made are highlighted in red in the revised version (included in full after the point-to-point responses to RC2) but most of them relate to the passage between lines 75 and 87:

[revised manuscript text omitted]

…Also, I feel that the source apportionment method is loosely defined in the methods session. Expanding the description of the method with an example of the mechanics would strengthen the paper.

This point was also raised by Reviewer #1 and expanded the description of the apportionment method to allow a better understanding of our methodology and results. In addition, we refer the reader to a new publication (Shu et al., 2023) that came available during the review process of our work. That paper, published in Geoscientific Model Development fully details the latest implementation of CMAQ-ISAM and provides a sample application, and compares results to other apportionment techniques.

The changes made in section 2.1 regarding model description are as follows:

[revised manuscript text omitted]

- Sillman, S.: The use of NOy, H₂O₂, and HNO₃ as indicators for ozone-NOx-hydrocarbon sensitivity in urban locations, Journal of Geophysical Research, 100, 14175-14188, https://doi.org/10.1029/94JD02953, 1995.

…It also feels that the discussion of the model evaluation has been skipped, and something is missing in the paper to convince the reader of the model's capabilities to reproduce the meteorology and chemical environment of the period study. It would be helpful to show a time series of O3 at some representative sites (urban, regional, and suburban) to show the diurnal and day-to-day variability (e.g., hourly ozone) and the model performance. This is important as it gives the reader a general idea of the O3 evolution and the pollution episodes, which are investigated throughout the study. Once these issues have been addressed, along with the points below, the manuscript will be suitable for publication.

Following the suggestions from both reviewers we included a new section (3.1) to provide a better view of the model performance assessment. We keep the detailed results of model assessment for each air quality monitoring station in the supplementary material (Table S3 in the revised version of our submission) because we think it helps the interpretation of site-specific results and complements the information given by aggregated statistics (Table 1). All statistics have been revised and harmonized. In addition to this, we added two new tables (Table S4 and Table S5) to illustrate the differences on model performance (both CMAQ and WRF) depending on the circulation pattern. Following the suggestion of Reviewer #2 we also show the comparison of observed and modeled O₃ series for 3 representative sites (pinpointed now in the revised version of Figure 1) as Figure 3. Besides illustrating the capabilities of the model and the reason for the statistical results obtained, it serves to present the features of the study period. Although a detail investigation of the causes for model discrepancies with observation is out of the scope of this contribution, we think it helps understanding potential reasons for performance differences found. As discussed in this new section, the difficulty of the meteorological model to reproduce wind fields under very weak forcing conditions (accumulation patterns) may contribute to the larger bias found in CMAQ outputs for that circulation type. We acknowledge this limitation and put our results in context with a critical discussion of our results and those from other relevant studies. We think this new section demonstrates a reasonable performance to study ground-level O₃. Furthermore, the results shown in Figure 8 (Figure 9 now) suggest a robust model performance also to describe O₃ mixing rations aloft. We think this is enough to build the confidence in the ability of the system to accurately describe ozone typical features and thus, we believe the modeling tool is fit for the purpose of the research at hand.

The new section 3.1 is as follows:

**3.1 Ozone levels during the study period and model evaluation**

While this period was hotter and dryer than most of recent summers, July 2016 may be representative of typical summer conditions in the Madrid region and included a concatenation of characteristic local circulation patterns (Plaza et al., 1997) with direct implications on ground-level O₃ (Querol et al., 2018; Escudero et al., 2019).  Figure 3 presents both observed and modeled

concentration series at representative points (Figure 1), and shows the venting and accumulation days identified in Querol et al., (2018). The time series demonstrate that $O_3$ levels are significantly lower under venting conditions, although significant differences are found depending on the location, which supports the need to use high-resolution modeling systems to analyze pollution dynamics in the Madrid region. On the other hand, accumulation patterns tend to produce higher concentrations (up to 175 μg/m$^3$), especially during July 27$^{th}$.

It can be observed that the model is able to reproduce the temporal patterns, as confirmed by the high correlation coefficients (r) and index of agreement (IOA) shown in Table 1. The statistical evaluation demonstrates a reasonable model performance, yielding better statistical results than recent simulation studies in this domain. Pay et al. (2019) reported an aggregated correlation coefficient of = 0.66 and mean bias (MB) of 22.5 ug/m$^3$ for the central region of the Iberian Peninsula. In this study, we obtained an average r value of 0.74 and a MB of 6.2 ug/m$^3$. Of note, 95.2% and 66.7% of the r values for the locations of the 42 monitoring stations used in this study are larger than 0.6 and 0.7, respectively while the overall normalized mean bias (NMB) is only 9 %. The results for a series of common statistics (Borge et al., 2010) for each of the monitoring sites in our modeling domain can be found in Table S3. The model, however, may have some difficulties capturing the amplitude of observed $O_3$ series and fails to accurately reproduce concentration peaks some days. This is evidenced by the relatively large error in comparison with the bias (23% and 9%, respectively as an average over the 42 monitoring stations in the modeling domain). In the supplementary material (Table S4), we present a separate model performance assessment for accumulation and advective patterns showing that the main differences among them relate to errors, both MGE and RMSE that are systematically higher for accumulation periods. This may be related to the limitations of the meteorological model to depict atmospheric circulation during stagnant conditions suggested by Pay et al., (2019). Even when WRF was found to outperform other models for this particular episode (Escuedero et al., 2019), the ability to reproduce wind direction and wind speed clearly deteriorates for accumulation periods, as shown in Table S5.,

As expected, results are poorer for urban background and traffic locations, since the typical spatio-temporal representativeness of the measurements in such locations is not comparable with that of a mesoscale modeling system, even with 1 km$^2$ spatial resolution."

[Figure]

**Figure. 3. Observed and predicted concentration series for selected locations (1-SMV: a rural location in the southwestern area of Madrid region, 2-ALG: a suburban location in the northeastern area of Madrid region and 3-RET: an urban background site in Madrid city center).**

And the new tables included in the supplementary material are:

**Table S4. Model performance statistics (dimensionless unless noted otherwise) by station type and circulation pattern for ground-level O₃ concentration.**

| Station | Pattern | n | FAC2 | MB (µgm⁻³) | MGE (µgm⁻³) | NMB | NMGE | RMSE (µgm⁻³) | r | IOA |
|---|---|---|---|---|---|---|---|---|---|---|
| Rural | Accumulation | 240 | 0.98 | -6.7 | 15.29 | -0.06 | 0.14 | 18.83 | 0.83 | 0.66 |
| | Advective | 232 | 0.98 | 3.1 | 9.31 | 0.04 | 0.11 | 12.97 | 0.83 | 0.73 |
| | Other | 3211 | 0.98 | -3.0 | 14.01 | -0.03 | 0.15 | 18.30 | 0.75 | 0.67 |
| Suburban | Accumulation | 474 | 0.96 | -4.8 | 20.24 | -0.05 | 0.20 | 26.69 | 0.76 | 0.68 |
| | Advective | 468 | 0.92 | 7.3 | 13.59 | 0.10 | 0.19 | 19.69 | 0.75 | 0.68 |
| | Other | 6412 | 0.94 | 2.6 | 17.18 | 0.03 | 0.20 | 23.22 | 0.73 | 0.68 |
| Urban background | Accumulation | 669 | 0.89 | 2.4 | 23.46 | 0.03 | 0.26 | 31.04 | 0.69 | 0.66 |
| | Advective | 670 | 0.89 | 11.4 | 16.95 | 0.17 | 0.25 | 22.34 | 0.72 | 0.60 |
| | Other | 9014 | 0.89 | 8.5 | 20.41 | 0.11 | 0.25 | 27.08 | 0.68 | 0.65 |
| Industrial | Accumulation | 96 | 0.95 | 4.7 | 16.40 | 0.05 | 0.18 | 20.15 | 0.86 | 0.73 |
| | Advective | 96 | 0.97 | 9.1 | 12.55 | 0.13 | 0.18 | 15.26 | 0.82 | 0.65 |
| | Other | 1278 | 0.95 | 7.9 | 14.54 | 0.10 | 0.18 | 18.79 | 0.83 | 0.71 |
| Urban traffic | Accumulation | 510 | 0.91 | 3.5 | 20.09 | 0.04 | 0.22 | 25.81 | 0.79 | 0.69 |
| | Advective | 522 | 0.87 | 15.8 | 18.22 | 0.25 | 0.28 | 24.55 | 0.69 | 0.55 |
| | Other | 7086 | 0.87 | 11.0 | 19.98 | 0.14 | 0.25 | 26.72 | 0.73 | 0.65 |

**Table S5. Model (WRF) performance statistics by circulation pattern for basic meteorological variables**

| Variable | Pattern | FAC2 | MB | MGE | NMB | NMGE | r | IOA |
|---|---|---|---|---|---|---|---|---|
| Temperature (T2) | Accumulation | 1.00 | -1.4 K | 2.0 K | -0.05 | 0.07 | 0.92 | 0.81 |
| | Advection | 1.00 | -0.5 K | 1.5 K | -0.02 | 0.06 | 0.96 | 0.86 |
| | Other | 1.00 | -0.8 K | 1.6 K | -0.03 | 0.06 | 0.96 | 0.85 |
| Wind speed (WS10) | Accumulation | 0.63 | 0.9 m/s | 1.7 m/s | 0.31 | 0.63 | 0.30 | 0.33 |
| | Advection | 0.78 | 0.7 m/s | 1.5 m/s | 0.17 | 0.37 | 0.59 | 0.55 |
| | Other | 0.71 | 0.5 m/s | 1.3 m/s | 0.18 | 0.46 | 0.58 | 0.55 |
| Wind direction | Accumulation | 0.61 | -34.3 ° | 90.7 ° | -0.24 | 0.63 | 0.26 | 0.55 |
| | Advection | 0.87 | 6.5 ° | 34.5 ° | 0.05 | 0.25 | 0.79 | 0.81 |
| | Other | 0.77 | -9.2 ° | 60.8 ° | -0.06 | 0.38 | 0.53 | 0.68 |

**General Comments**

When you say that the contribution from biogenic emissions is relatively small and therefore excluded from the analysis, are you referring to the contribution to regional scale O3? Could you consider including an evaluation of biogenic VOCs, e.g., a time series comparison of isoprene or a statement regarding the performance of the MEGAN model? Biogenic VOCs, in particular isoprene, are important contributors to O3 formation during photochemical O3 episodes, particularly in rural areas but also in urban environments (Dunker et al., 2016), due to their reactivity and abundance. You concluded that biogenic sources are responsible for 42.4% of the total VOCs domain-wide, so an important impact from this source should be expected.

We reformulated the narrative not to underestimate the influence of biogenic VOCs (BVOC) and compare our results with the findings of previous studies in the literature. We make clear that direct

comparison is not possible since the interpretation depends on the specific source apportionment methodology used and the specific model domain and scale of application.

Considering that caveat, we found that global and continental studies usually attribute a more important role to biogenic volatile organic compounds (BVOC) in the explanation of $O_3$ budgets. Nonetheless, our results seem consistent for other studies in the Iberian Peninsula or the Madrid region specifically. Nonetheless, we highlight the need to use caution comparing results from different studies, so the reader is not misled by methodological differences. We connect this discussion with the reactivity of specific VOCs for better understanding of the impact of VOCs on ground-level $O_3$. Following Reviewer #2's suggestion, we focus on isoprene to illustrate the question. Unfortunately, there are no routine measurements of isoprene in the Madrid region, but some measurements were performed during the experimental campaign of Querol et al., (2018) that are compared with our model predictions in Figure AC2 below (included in the supplementary material of our revised submission as Figure S3).

[Figure]

**Figure AC2. Comparison of isoprene ground-level mixing ratios predicted by CMAQ (left) and measurements made in Majadahonda (suburban site) by Querol et al., (2018) (right). Both graphs present the hourly values during the day averaged over the period July 5th and July 19th. The source of the right-hand panel is Pérez et al., (2016).**

Reference (added to the supplementary material):

- Pérez, N., A. Alastuey, C. Reche, M. Ealo, G. Titos, A. Ripoll, M.C. Minguillón, F. J. Gómez-Moreno, E. Alonso-Blanco, E. Coz, E. Díaz, B. Artíñano, S. García dos Santos, R. Fernández-Patier, A. Saiz-López, F. Serranía, M. Anguas-Ballesteros, B. TemimeRoussel, N. Marchand, D. C. S. Beddows, R. M. Harrison y X. Querol. Campaña intensiva de medidas de UFP, O3 y sus precursores en el área de Madrid: medidas en superficie., https://www.miteco.gob.es/content/dam/miteco/es/calidad-y-evaluacion-ambiental/temas/atmosfera-y-calidad-del-aire/anexo_informea33_madrid_tcm30-561368.pdf (last access: [January 22, 2004]), 2016.

Although the temporal pattern of isoprene was acceptably reproduced by CMAQ, the mixing ratio of this specie was underestimated by nearly a factor of 2 during the period where measurements were available. While this may be a potential factor to explain the relatively low impact, the literature suggests that other anthropogenic VOC may have a larger ozone formation potential

(OFP). More importantly, our results suggest that the contribution of all anthropogenic local emissions are more relevant than those of BVOC because O3 formation in this study was mainly driven by NOX emissions. The discussion about biogenics is included at the beginning of the revised section 3.2 (lines 297-305) and it reads as it follows:

[revised manuscript text omitted]

Evaluation of the model is reported in the supplemental material and loosely mentioned in the manuscript. Please adjust section 3 and provide a quantitative statement of the model performance for meteorology and chemistry (especially O3) for the model domains, along with some plots. This could be a spatial contour plot showing the model and observed mean or P95 of O3 or time series of O3 at some representative sites.

Following the suggestion from Reviewer #2, we included a new subsection (3.1) to provide a better view of model performance. Please, see the response to a previous question related to this and the changes made.

**Specific Comments**

Line 32:33: 'These measures, however, have failed to significantly improve ozone (O3) ambient concentration levels'. I feel this is a strong affirmation that needs to be re-phrased, taking into account the nonlinearity nature of O3 formation and the different aspects related to the concentrations observed in different chemical environments (e.g., urban, rural, and suburban) as well as the effects of emissions reductions such as the urban decrement.

Considering the point raised, we revised the sentence that now reads: "These measures, however, have not reported comparable reductions of ozone (O₃) ambient concentration levels."
We don't further elaborate on this because we feel it would be redundant with the following discussion in the introduction (lines 58-69 of the revised manuscript) about recent ozone trends.

Line 140: It would be helpful to have a paragraph describing how experiments were designed, for instance, how the chemical cycling is performed and how often the meteorology is restarted.
We provided additional information on the initialization of the meteorological model (WRF) in the supplementary material to complement the information about model setup and modeling spatial and temporal domains. Additionally, we amended Table S1 since it didn't reflect the actual setup used in this experiment. Now the material in the supplement regarding WRF is as follows:

**Table S1. WRF model physics options and parametrizations.**

| Option | Setup |
| --- | --- |
| Initialization | GFS |
| Shortwave radiation | Dudhia scheme |
| Longwave radiation | GFDL |
| Land-surface model | Noah LSM |
| Microphysics scheme | WSM 6-class Graupel scheme |
| PBL Scheme | YSU scheme |
| Surface Layer option | Monin-Obukhov |
| Cumulus Parametrization | No |
|  |  |
| Nudging | Yes |

The WRF model was initialized from global reanalysis made available by NCEP (National Centers for Environmental Prediction) from outputs of the GFS (Global Forecast System) (ds083.0). They have a spatial resolution of 1º x 1º and a temporal resolution of 6 hours ((00Z, 06Z, 12Z, 18Z). Data assimilation was applied (via nudging excluding the planetary boundary layer) for a more realistic representation of meteorological fields using both, surface observations from NCEP ADP Global Surface Observational Weather Data (ds461.0) and vertical soundings from NCEP ADP Global Upper Air Observational Weather Data (ds351.0).

Line 249: The link provided does not work
Line 360: The link provided does not work

Thanks for letting us know. We have revised all links, including those of the references both new and those already included in the original manuscript.

[revised manuscript text omitted]

The WRF model was initialized from global reanalysis made available by NCEP (National Centers for Environmental Prediction) from outputs of the GFS (Global Forecast System) (ds083.0). They have a spatial resolution of 1º x 1º and a temporal resolution of 6 hours ((00Z, 06Z, 12Z, 18Z). Data assimilation was applied (via nudging excluding the planetary boundary layer) for a more realistic representation of meteorological fields using both, surface observations from NCEP ADP Global Surface Observational Weather Data (ds461.0) and vertical soundings from NCEP ADP Global Upper Air Observational Weather Data (ds351.0).

**Table S2. Horizontal dimensions and resolution of WRF and CMAQ modeling domains.**

| Domains | Geographic area | WRF X-Y dimensions (grid cells) | CMAQ X-Y dimensions (grid cells) | Horizontal resolution (km) |
|---|---|---|---|---|
| D1 | Europe | 560 x 496 | 459 x 406 | 12 |
| D2 | Iberian Peninsula | 384 x 312 | 300 x 240 | 4 |
| D3 | Greater Madrid area | 256 x 256 | 136 x 144 | 1 |

Table S3. Compliance objectives of Directive 2008/50/EC and those defined by the World Health Organization (WHO) in relation to the pollutant analyzed.

| Average time | O₃ (μg m⁻³) | |
|---|---|---|
| | R.D 102/2011 | OMS |
| Maximum daily eight-hour mean | 120 μg m⁻³ not to be exceeded on more than 25 days per calendar year averaged over three years | 100 μg m⁻³ 3-4 exceedance days per year |

**Table S3. Model performance statistics (dimensionless unless noted otherwise) by station for ground-level O₃ concentration.**

| STATION | TYPE | FAC2 | MB ($\mu gm^{-3}$) | MGE ($\mu gm^{-3}$) | NMB | NMGE | RMSE ($\mu gm^{-3}$) | r | IOA |
|---|---|---|---|---|---|---|---|---|---|
| Arganda del Rey | Industrial | 0.96 | 5.8 | 14.6 | 0.07 | 0.17 | 18.6 | 0.84 | 0.72 |
| Fuenlabrada | Industrial | 0.95 | 9.8 | 14.4 | 0.13 | 0.19 | 18.7 | 0.84 | 0.69 |
| Villa del Prado | Rural | 0.99 | -0.8 | 11.8 | -0.01 | 0.13 | 15.3 | 0.82 | 0.72 |
| S.Mde Valdeiglesias | Rural | 1.00 | 0.0 | 10.5 | 0.00 | 0.11 | 13.7 | 0.80 | 0.71 |
| Orusco de Tajuña | Rural | 1.00 | -10.0 | 12.7 | -0.10 | 0.12 | 16.1 | 0.84 | 0.66 |
| Guadalix de la sierra | Rural | 0.92 | 7.6 | 17.9 | 0.09 | 0.21 | 22.7 | 0.79 | 0.67 |
| El Atazar | Rural | 0.99 | -11.2 | 16.1 | -0.11 | 0.15 | 20.7 | 0.69 | 0.58 |
| Algete | Suburban | 1.00 | -4.4 | 13.1 | -0.05 | 0.14 | 17.1 | 0.81 | 0.72 |
| La Sagra | Suburban | 0.94 | 7.3 | 14.5 | 0.09 | 0.18 | 20.0 | 0.81 | 0.71 |
| Mostoles | Suburban | 0.94 | 8.0 | 15.5 | 0.10 | 0.19 | 20.6 | 0.83 | 0.71 |
| Majadahonda | Suburban | 0.96 | -2.7 | 15.7 | -0.03 | 0.17 | 21.4 | 0.81 | 0.71 |
| Valdemoro | Suburban | 0.91 | 6.7 | 16.0 | 0.08 | 0.19 | 22.1 | 0.80 | 0.71 |
| Rivas Vaciamadrid | Suburban | 0.91 | 7.0 | 17.3 | 0.09 | 0.21 | 23.0 | 0.81 | 0.70 |
| Torrejon de Ardoz | Suburban | 0.90 | 10.1 | 17.6 | 0.13 | 0.22 | 23.7 | 0.82 | 0.70 |
| Azuqu. de Henares | Suburban | 0.95 | 3.6 | 16.8 | 0.04 | 0.20 | 21.6 | 0.78 | 0.70 |
| Toledo2 | Suburban | 0.95 | -0.9 | 16.3 | -0.01 | 0.18 | 22.0 | 0.72 | 0.68 |
| Aranjuez | Suburban | 0.91 | 9.3 | 16.7 | 0.11 | 0.20 | 22.5 | 0.77 | 0.67 |
| El Pardo | Suburban | 0.92 | -0.2 | 22.2 | 0.00 | 0.24 | 28.0 | 0.74 | 0.65 |
| Casa de campo | Suburban | 0.94 | 1.7 | 20.1 | 0.02 | 0.23 | 26.7 | 0.61 | 0.63 |
| Juan Carlos I | Suburban | 0.90 | -4.6 | 24.2 | -0.05 | 0.27 | 31.0 | 0.61 | 0.63 |
| Alcorcón | Urb.Background | 0.96 | 4.8 | 14.6 | 0.06 | 0.18 | 19.7 | 0.83 | 0.73 |
| Guadalajara | Urb.Background | 0.94 | 7.4 | 15.6 | 0.09 | 0.19 | 21.4 | 0.77 | 0.70 |
| Tres olivos | Urb.Background | 0.92 | -4.0 | 22.8 | -0.04 | 0.25 | 29.2 | 0.66 | 0.63 |
| Villaverde | Urb.Background | 0.86 | 13.1 | 22.2 | 0.17 | 0.29 | 29.4 | 0.66 | 0.61 |
| Farolillo | Urb.Background | 0.88 | 5.8 | 22.4 | 0.07 | 0.27 | 29.7 | 0.62 | 0.62 |
| Retiro | Urb.Background | 0.86 | 11.9 | 23.0 | 0.16 | 0.31 | 29.3 | 0.64 | 0.60 |
| Barajas pueblo | Urb.Background | 0.81 | 11.2 | 25.1 | 0.15 | 0.33 | 32.2 | 0.65 | 0.62 |
| Arturo Soria | Urb.Background | 0.84 | 15.4 | 23.2 | 0.22 | 0.32 | 29.8 | 0.63 | 0.57 |
| Ench de Vallecas | Urb.Background | 0.88 | 6.2 | 21.1 | 0.07 | 0.25 | 27.9 | 0.66 | 0.64 |
| Plaza del Carmen | Urb.Background | 0.72 | 23.9 | 29.9 | 0.39 | 0.48 | 37.0 | 0.59 | 0.47 |
| Segovia 2 | Traffic | 0.97 | 3.5 | 13.6 | 0.04 | 0.16 | 17.0 | 0.84 | 0.71 |
| Vill.de Salvanés | Traffic | 0.99 | 3.1 | 10.6 | 0.04 | 0.12 | 14.6 | 0.78 | 0.71 |
| Colmenar Viejo | Traffic | 0.99 | -0.7 | 13.0 | -0.01 | 0.14 | 17.3 | 0.78 | 0.69 |
| Alcobendas | Traffic | 0.93 | -0.8 | 17.8 | -0.01 | 0.20 | 23.6 | 0.80 | 0.70 |
| Getafe | Traffic | 0.92 | 8.8 | 16.8 | 0.11 | 0.21 | 23.1 | 0.80 | 0.70 |
| Alcala de Henares | Traffic | 0.87 | 10.0 | 19.3 | 0.13 | 0.24 | 25.0 | 0.83 | 0.69 |
| Leganes | Traffic | 0.87 | 12.4 | 18.4 | 0.16 | 0.24 | 25.7 | 0.79 | 0.67 |
| Barrio del Pilar | Traffic | 0.88 | 9.0 | 20.8 | 0.11 | 0.27 | 28.2 | 0.64 | 0.62 |
| Coslada | Traffic | 0.79 | 18.9 | 24.7 | 0.27 | 0.35 | 31.2 | 0.80 | 0.61 |
| Collado Villalba | Traffic | 0.78 | 19.3 | 23.6 | 0.26 | 0.32 | 31.7 | 0.73 | 0.59 |
| Escuelas Aguirre | Traffic | 0.83 | 16.5 | 23.7 | 0.24 | 0.35 | 29.9 | 0.63 | 0.54 |
| Pzs. Fedz Ladreda | Traffic | 0.80 | 22.1 | 26.7 | 0.34 | 0.41 | 33.4 | 0.6 | 0.5 |

**Table S4. Model performance statistics (dimensionless unless noted otherwise) by station type and circulation pattern for ground-level $O_3$ concentration.**

| Station | Pattern | n | FAC2 | MB ($\mu gm^{-3}$) | MGE ($\mu gm^{-3}$) | NMB | NMGE | RMSE ($\mu gm^{-3}$) | r | IOA |
|---|---|---|---|---|---|---|---|---|---|---|
| Rural | Accumulation | 240 | 0.98 | -6.7 | 15.29 | -0.06 | 0.14 | 18.83 | 0.83 | 0.66 |
| | Advective | 232 | 0.98 | 3.1 | 9.31 | 0.04 | 0.11 | 12.97 | 0.83 | 0.73 |
| | Other | 3211 | 0.98 | -3.0 | 14.01 | -0.03 | 0.15 | 18.30 | 0.75 | 0.67 |
| Suburban | Accumulation | 474 | 0.96 | -4.8 | 20.24 | -0.05 | 0.20 | 26.69 | 0.76 | 0.68 |
| | Advective | 468 | 0.92 | 7.3 | 13.59 | 0.10 | 0.19 | 19.69 | 0.75 | 0.68 |
| | Other | 6412 | 0.94 | 2.6 | 17.18 | 0.03 | 0.20 | 23.22 | 0.73 | 0.68 |
| Urban background | Accumulation | 669 | 0.89 | 2.4 | 23.46 | 0.03 | 0.26 | 31.04 | 0.69 | 0.66 |
| | Advective | 670 | 0.89 | 11.4 | 16.95 | 0.17 | 0.25 | 22.34 | 0.72 | 0.60 |
| | Other | 9014 | 0.89 | 8.5 | 20.41 | 0.11 | 0.25 | 27.08 | 0.68 | 0.65 |
| Industrial | Accumulation | 96 | 0.95 | 4.7 | 16.40 | 0.05 | 0.18 | 20.15 | 0.86 | 0.73 |
| | Advective | 96 | 0.97 | 9.1 | 12.55 | 0.13 | 0.18 | 15.26 | 0.82 | 0.65 |
| | Other | 1278 | 0.95 | 7.9 | 14.54 | 0.10 | 0.18 | 18.79 | 0.83 | 0.71 |
| Urban traffic | Accumulation | 510 | 0.91 | 3.5 | 20.09 | 0.04 | 0.22 | 25.81 | 0.79 | 0.69 |
| | Advective | 522 | 0.87 | 15.8 | 18.22 | 0.25 | 0.28 | 24.55 | 0.69 | 0.55 |
| | Other | 7086 | 0.87 | 11.0 | 19.98 | 0.14 | 0.25 | 26.72 | 0.73 | 0.65 |

**Table S5. Model (WRF) performance statistics by circulation pattern for basic meteorological variables**

| Variable | Pattern | FAC2 | MB | MGE | NMB | NMGE | r | IOA |
|---|---|---|---|---|---|---|---|---|
| Temperature (T2) | Accumulation | 1.00 | -1.4 K | 2.0 K | -0.05 | 0.07 | 0.92 | 0.81 |
| | Advection | 1.00 | -0.5 K | 1.5 K | -0.02 | 0.06 | 0.96 | 0.86 |
| | Other | 1.00 | -0.8 K | 1.6 K | -0.03 | 0.06 | 0.96 | 0.85 |
| Wind speed (WS10) | Accumulation | 0.63 | 0.9 m/s | 1.7 m/s | 0.31 | 0.63 | 0.30 | 0.33 |
| | Advection | 0.78 | 0.7 m/s | 1.5 m/s | 0.17 | 0.37 | 0.59 | 0.55 |
| | Other | 0.71 | 0.5 m/s | 1.3 m/s | 0.18 | 0.46 | 0.58 | 0.55 |
| Wind direction | Accumulation | 0.61 | -34.3 ° | 90.7 ° | -0.24 | 0.63 | 0.26 | 0.55 |
| | Advection | 0.87 | 6.5 ° | 34.5 ° | 0.05 | 0.25 | 0.79 | 0.81 |
| | Other | 0.77 | -9.2 ° | 60.8 ° | -0.06 | 0.38 | 0.53 | 0.68 |

[Figure]

**Figure S1. Spatially-averaged source apportionment (%) over the whole Madrid Region for (a) O₃ monthly mean and (b) 90ᵗʰ 1-hour percentile, including the sectoral breakdown within anthropogenic contributions.**

[Figure]

**Figure S2. Absolute contribution (ppb) to the monthly mean 1-hour 90ᵗʰ O₃ percentile of the SNAP 06 sector (use of solvents and other products) and biogenic emissions.**

[Figure]

**Figure S3. Comparison of isoprene ground-level mixing ratios predicted by CMAQ (left) and measurements made in Majadahonda (suburban site) by Querol et al., (2018) (right). Both graphs present the hourly values during the day averaged over the period July 5th and July 19th. The source of the right-hand panel is Pérez et al., (2016).**

Reference:

- Pérez, N., A. Alastuey, C. Reche, M. Ealo, G. Titos, A. Ripoll, M.C. Minguillón, F. J. Gómez-Moreno, E. Alonso-Blanco, E. Coz, E. Díaz, B. Artíñano, S. García dos Santos, R. Fernández-Patier, A. Saiz-López, F. Serranía, M. Anguas-Ballesteros, B. TemimeRoussel, N. Marchand, D. C. S. Beddows, R. M. Harrison y X. Querol. Campaña intensiva de medidas de UFP, O3 y sus precursores en el área de Madrid: medidas en superficie., https://www.miteco.gob.es/content/dam/miteco/es/calidad-y-evaluacion-ambiental/temas/atmosfera-y-calidad-del-aire/anexo_informea33_madrid_tcm30-561368.pdf (last access: [January 22, 2004]), 2016.

[Figure]

**Figure S4. Maximum 1-hour attribution of stratospheric transport (PVO3) to ground level**

[Figure]

**Figure S5.** Absolute contribution to the **1-hour 90th O₃ percentile** of the main emitting sectors.

[Figure]

**Figure S6. Accumulation period: evolution during July 27th. From left to right, plan view and SE-NW cross section (up to 5 km height) O₃ mixing ratios (ppb), NOx (ppb) and VOCs (ppb) at 3:00, 9:00; 15:00, 21: 00 UTC hours. MD = Madrid City.**

[Figure]

**Figure S7. Accumulation period: evolution during July 6th. From left to right, plan view and NE-SW cross section (up to 5 km height) O₃ mixing ratios (ppb), NOx (ppb) and VOCs (ppb) at 3:00, 9:00; 15:00, 21: 00 UTC hours. MD = Madrid City.**

[Figure]

**Figure S8.** Accumulation period: **evolution** during July 6th. From left to right, plan view and SE-NW cross section (up to 5 km height) O₃ **mixing ratios** (ppb), NOx (ppb) and VOCs (ppb) at 3:00, 9:00; 15:00, 21: 00 UTC hours. MD = Madrid City.

[Figure]

**Figure S9.** Advection period: **evolution** during July 13th. From left to right, plan view and SE-NW cross section (up to 5 km height) O₃ **mixing ratios** (ppb), NOx (ppb) and VOCs (ppb) at 3:00, 9:00; 15:00, 21: 00 UTC hours. MD = Madrid City.

[Figure]

**Figure S10. Advection period: evolution during July 20th. From left to right, plan view and NE-SW cross section (up to 5 km height) O₃ mixing ratios (ppb), NOx (ppb) and VOCs (ppb) at 3:00, 9:00; 15:00, 21: 00 UTC hours. MD = Madrid City.**

[Figure]

**Figure S11.** Advection period: **evolution** during July 20th. From left to right, plan view and SE-NW cross section (up to 5 km height) O₃ **mixing ratios** (ppb), NOx (ppb) and VOCs (ppb) at 3:00, 9:00; 15:00, 21: 00 UTC hours. MD = Madrid City.

[Figure]

**Figure S12. O₃ mixing ratios** (ppb) at 09:00 UTC for July 27th (accumulation period) and July 13th (advective period). From left to right, plan view, NE-SW and SE-NW cross sections (up to 5 km height). MD = Madrid City.

[Figure]

**Figure S13.** Geographical division (quadrants) of the study area for the analysis of individual monitoring station locations

[Figure]

**Figure S14. Hourly contribution (ppb) for the monthly average at the location of monitoring sites by geographical quadrant.**

[Figure]

**Figure S15. Hourly contribution (ppb) for July 27th, 2016 at the location of monitoring sites by geographical quadrant.**

[Figure]

**Figure S16.** Hourly contribution (ppb) for July 13th, 2016 at the location of monitoring sites by geographical quadrant.

**Table S4. Model performance statistics by station type for ground-level O3 concentration.**

| Station | FAC2 | MB (µg m-3) | MGE (µg m-3) | NMB | NMGE | RMSE (µg m-3) | r | IOA |
|---|---|---|---|---|---|---|---|---|
| Industrial | 0.95 | 7.8 | 14.5 | 0.10 | 0.18 | 18.7 | 0.84 | 0.71 |
| Rural | 0.98 | -2.9 | 13.8 | -0.03 | 0.14 | 18.1 | 0.76 | 0.68 |
| Suburban | 0.94 | 2.4 | 17.15 | 0.03 | 0.20 | 23.3 | 0.74 | 0.69 |
| Urban background | 0.89 | 8.3 | 20.4 | 0.10 | 0.25 | 27.1 | 0.69 | 0.65 |
| Urban traffic | 0.88 | 10.8 | 19.9 | 0.14 | 0.25 | 26.5 | 0.73 | 0.65 |

Table S4. Model performance statistics by station type for ground-level O3 concentration.

---

## Author Response (AR2)

**Response to Editor decision (publish subject to minor revisions)**

Dear editor, thank you for the opportunity to refine our submission according to the points raised by both reviewers in the second revision round. Once more, we acknowledge them for the time devoted to our manuscript and for fruitful suggestions that definitively helped us to improve our work.

In the document attached we provide point-to-point responses to each of their additional questions and comments for your final consideration. Consistently with the response to reviewers previously submitted, the author comments are structured according to ACP guidelines and follow the recommended sequence: comments from the referees (report #2 and report #1) are shown in blue, then we provide our responses in black and the corresponding changes made in the manuscript and/or supplementary material (over the corresponding R1 versions) are included in red.
* * *
**Responses to report #2 (Anonymous referee #1)**

The authors have done a good job in revising the manuscript. Before I can recommend the manuscript for publication, however, a few details still need to be clarified:

Thank you for your contributions to improve our manuscript and for further recommendations to clarify remaining minor issues.

1) Figure 2: Please add the unit (%) to the axis. Moreover, I wonder if it would be much more helpful for the reader if the figure would show the share of the emissions during the analyzed period and not for the whole year (the share of biogenic emissions might be much larger during summer as for the annual mean?).

Yes, we agree that is a good point. Now Figure 2 refers exclusively to the temporal span of the modeling exercise (July 2016). We have added the % to the x axis and removed the description of the tagged sectors since it is now shown in Table 1, according to the following suggestion from Reviewer #1.

While emissions from vegetation (SNAP 10 and SNAP 11 sectors) represent 42.2% of total VOC emissions on an annual basis, this share raises to 72.2% specifically for this summer month.

Of note, in the process of double-checking emission figures we found an error regarding $NO_X$ emissions from soils, included in this tag. While emissions inputs are correct, they were mistakenly reported in Figure AC6 within the responses to reviewers during the first revision of our manuscript. They are relatively small (around 224 t in this period, distributed as shown in Figure AC1 below), but by no means negligible. They represent 3.8% of total $NO_X$ emissions, which is very similar to the estimate of MEGAN for the whole Europe according to Visser et al. (2019). These authors, however, point out that actual $NO_X$ emissions from agricultural soils may be larger. That would be consistent with other estimates at global scale that suggest that soils may be responsible for up to 15% of total $NO_X$ emissions (Weng et al., 2020). According to Lu et al. (2021), $NO_X$ emissions from agricultural

soils may be a significant limitation towards meeting $O_3$ standards since they significantly reduce the sensitivity of ozone to anthropogenic emissions. This specific issue may be addressed by further future specific studies to clarify to what extent this source may be diminishing the potential of anthropogenic emission abatement strategies.

[Figure]

**Figure AC1. Soil NO$_X$ emissions (t) estimated in our modeling domain for July 2016**

We revised section 2.4 of our manuscript to adjust the discussion to monthly percentages and to correct the discussion regarding NO$_X$ emissions from soils. We included Table 1 for the sake of clarity and revised the manuscript for a consistent use of the abbreviations defined in the 3$^{rd}$ column to refer to tagged ozone sources. Section 2.4 now reads (changes in red):

**"2.4. Emission sources for the apportionment analysis**

[revised manuscript text omitted]

References added:

- Visser et al. (2019): https://doi.org/10.5194/acp-19-11821-2019

2) To my opinion a table which lists all the tagged sectors including a short description would be very helpful. Figure 2 lists only the emissions, Fig. 12 lists all tagged sectors but using acronyms only i.e. the reader has to search for the meaning of "OTH" or "PVO3". Also the tagged biogenic emissions are labeled SNAP10-11 in Figure 2 and BIO in Figure 12. Please use a unique naming scheme.

We made the necessary changes to incorporate this suggestion as discussed in the previous response. Figure 12, Figure S1, Figure S2, Figure S14, Figure S15 and Figure S16 have been modified by following this unique naming scheme.

3) I think the newly added discussion on the contribution of biogenic emissions in Sect. 3.2 is interesting, but the section seems unbalanced to me. The authors write: "O3 apportionment to biogenic emissions is not considered in Figure 4 because i) they have less interest from the point of view of possible abatement measures (Oliveira et al., 2023) and ii) their contribution is relatively

small". After this statement they add one page of discussion on biogenic emissions and only after this discussion the figure is discussed further. I feel this newly added part would much better fit into a dedicated "Discussion" section (or at the end of Sect. 3.2)

We concur with reviewer #1. We adopted a better-balanced structure where section 3.2 is split in two subsections (3.2.1. Non-anthropogenic sources and 3.2.2. Anthropogenic sources) with the following brief introduction:

"3.2. Spatial analysis of the source apportionment assessment

In this section, we discuss the contribution to ground-level $O_3$ of the tagged sources (Table 1) both, for monthly average and high values (illustrated by the 90th percentile, hereinafter P90). $O_3$ apportionment focusses on anthropogenic sources since they have more interest from the point of view of possible abatement measures (Oliveira et al., 2023) and have a larger contribution than that of SNAP10-11 (below 4% to total $O_3$ levels in this period). However, it is not a negligible apportionment since these groups account for 27% (monthly mean) and 22% (P90) of total $O_3$ averaged over the Madrid region when BC and IC are not considered (Figure S1). Non-anthropogenic emissions have been reported to play an important role on atmospheric photochemistry and they interact with manmade emissions so, they need to be considered in the process of designing policies to reduce tropospheric $O_3$ levels. Therefore, we discuss the potential role of emissions from agriculture and nature as well."

Furthermore, we added a brief discussion on the potential effects of $NO_x$ emissions from agricultural and forest soils within new subsection 3.2.1 (lines 289-293 of the revised manuscript):

"Of note, SNAP 10-11 include $NO_X$ emissions from soils (see section 2.4). Although they represent less than 4% of total $NO_X$ emissions in the domain, they may be underestimated by MEGAN (Visser et al., 2019). According to other studies, i.a. Weng et al. (2020), emissions from agricultural soils may be substantially higher and could pose a significant constrain towards the control of $O_3$ levels (Lu et al., 2021). Methods to reduce the uncertainty of $NO_X$ emissions estimates from soils as well as their role for $O_3$ control policies specifically for this region may be addressed in future research."

We added the later to the potential future research lines in the last paragraph of the conclusions section (lines 490-491):

"…speciation for specific sources. Furthermore, the role of biogenic $NO_X$ and VOC emissions may be further studied to understand the implications for $O_3$ control strategies in the Madrid region."

References added:

- Visser et al. (2019): https://doi.org/10.5194/acp-19-11821-2019
- Weng et al. (2020): https://doi.org/10.1038/s41597-020-0488-5
- Lu et al. (2021): https://doi.org/10.1038/s41467-021-25147-9, 2021

4) Please clarify L426: "This relates to second-order interactions between sources (U.S. EPA, 2022). This represents a negligible fraction in this study, i.e. ISAM could attribute the virtual totality of O3 to any of the other sources" What is meant by " second-order interactions between sources"?

We appreciate the reviewer for pointing out these lines of text. We realize that the term "second-order" has specific mathematical meaning and could lead the reader to confusion. What we mean here is that this contribution is negligible and minor. We also provide a reference where the reader can find more information about this issue. In our opinion, it would not be prudent to distract the reader from the main points of our work here with the lengthy explanation about this while another reference is available. Taking this suggestion into considerations, we made the following changes to the manuscript (lines 411-414):

"Although 100% of emitting sectors have been tagged, Figure 12 shows as well the contribution from "others" (OTH). This contribution is typically negligible and relates to minor model interactions between sources and species not considered by the ISAM model. Details are fully explained in the documents provided with the model release (U.S. EPA, 2022)."

5) Please fix wrong usage of citations with (author) instead of author (e.g. (Paoletti et al., 2014) )

We have carefully checked the reference style and made 12 corrections, including Paoletti et al. (2014).
* * *
**Responses to report #1 (Referee #2: Johana Romero Alvarez)**

The authors addressed the previously suggested comments, enhancing the manuscript's contribution to understanding the role of different emission sources in ozone formation. To improve the clarity of the discussion, I recommend the following specific revisions:

Thank you for the time devoted to our manuscript and for making additional suggestions to polish our manuscript.

Typographical and Citation Format Consistency:
I recommend a comprehensive review of the manuscript to check for typographical errors such as missing commas or periods in citations (e.g., Collet et al., (2018), should be Collet et al. (2018))

Thank for spotting that typo. We have carefully checked the reference style and made 12 corrections, including Collet et al. (2018).

Clarification on the Significance of Biogenic Emissions (Line 260):
The manuscript states that biogenic emissions account for 27% (monthly mean) and 22% (P90) of total O3 but are described as not negligible. To strengthen the narrative, it would be beneficial to elaborate on why these percentages, despite being lower than anthropogenic contributions, hold significant implications for ozone formation strategies and understanding.

Following reviewer #1's recommendation, we split section in two subsections dedicating one of them to the discussion of biogenic emissions (3.2.1. Non-anthropogenic sources). Now, we include a brief introduction in lines 247-254 (reproduced below) where we make this point explicit. We believe this helps to strengthen the narrative and makes clear the rationale to specifically discuss the role of emissions from agriculture and nature (SNAP 10-11).

"3.2. Spatial analysis of the source apportionment assessment

In this section, we discuss the contribution to ground-level $O_3$ of the tagged sources (Table 1) both, for monthly average and high values (illustrated by the 90th percentile, hereinafter P90). $O_3$ apportionment focusses on anthropogenic sources since they have more interest from the point of view of possible abatement measures (Oliveira et al., 2023) and have a larger contribution than that of SNAP10-11 (below 4% to total $O_3$ levels in this period). However, it is not a negligible apportionment since these groups account for 27% (monthly mean) and 22% (P90) of total $O_3$ averaged over the Madrid region when BC and IC are not considered (Figure S1). Non-anthropogenic emissions have been reported to play an important role on atmospheric photochemistry and they interact with manmade emissions so, they need to be considered in the process of designing policies to reduce tropospheric $O_3$ levels. Therefore, we discuss the potential role of emissions from agriculture and nature as well."

In addition, we reflected the need to further understand the role of non-anthropogenic emissions as a future research lines in the last paragraph of the conclusions section (lines 494-495):

"…speciation for specific sources. Furthermore, the role of biogenic $NO_X$ and VOC emissions may be further studied to understand the implications for $O_3$ control strategies in the Madrid region."

Comparison with Previous Studies (Line 263):
When discussing the comparison of your findings with those of previous studies (e.g., Sartelet et al., 2012), it would be beneficial to briefly outline the methodologies or data employed in these referenced studies, especially if they diverge from your approach. This detail will provide readers with a clearer understanding of the reasons behind any similarities or discrepancies in findings, enriching the discussion.

We acknowledge the suggestion although we feel that discussing the methodological differences with previous studies regarding biogenic contributions would be out of the scope of this paper. Nonetheless, we summarize the main methodological features that may lead to discrepancies within the new subsection 3.2.1. (lines 266-269):

"… It should be noted that different experimental design and apportionment algorithms would lead to significant differences (Zhang et al., 2017; Borge et al., 2022) preventing the direct comparison of the results from different studies. In addition to the apportionment methodology itself, the results may differ depending on the emission inventory used, the modeling scale and resolution, temporal span and sources tagging scheme."